# Rbpj expression in regulatory T cells is critical for restraining $T_H2$ responses

Michael Delacher [1,2,3], Christian Schmidl [2], Yonatan Herzig[4], Minka Breloer [5], Wiebke Hartmann[5], Fabian Brunk[6], Danny Kägebein[3], Ulrike Träger[3], Ann-Cathrin Hofer[3], Sebastian Bittner[1,2], Dieter Weichenhan[7], Charles D. Imbusch [8], Agnes Hotz-Wagenblatt[9], Thomas Hielscher[10], Achim Breiling [11], Giuseppina Federico[12], Hermann-Josef Gröne[12], Roland M. Schmid[13], Michael Rehli [2,14], Jakub Abramson[4] & Markus Feuerer [1,2,3]

The transcriptional regulator Rbpj is involved in T-helper ($T_H$) subset polarization, but its function in $T_{reg}$ cells remains unclear. Here we show that $T_{reg}$-specific Rbpj deletion leads to splenomegaly and lymphadenopathy despite increased numbers of $T_{reg}$ cells with a polyclonal TCR repertoire. A specific defect of Rbpj-deficient $T_{reg}$ cells in controlling $T_H2$ polarization and B cell responses is observed, leading to the spontaneous formation of germinal centers and a $T_H2$-associated immunoglobulin class switch. The observed phenotype is environment-dependent and can be induced by infection with parasitic nematodes. Rbpj-deficient $T_{reg}$ cells adopt open chromatin landscapes and gene expression profiles reminiscent of tissue-derived $T_H2$-polarized $T_{reg}$ cells, with a prevailing signature of the transcription factor Gata-3. Taken together, our study suggests that $T_{reg}$ cells require Rbpj to specifically restrain $T_H2$ responses, including their own excessive $T_H2$-like differentiation potential.

[1] Chair for Immunology, University Regensburg and University Hospital Regensburg, Franz-Josef-Strauss-Allee 11, 93053 Regensburg, Germany. [2] Regensburg Center for Interventional Immunology (RCI), University Regensburg and University Hospital Regensburg, Franz-Josef-Strauss-Allee 11, 93053 Regensburg, Germany. [3] Immune Tolerance Group, Tumor Immunology Program, German Cancer Research Center (DKFZ), Im Neuenheimer Feld 280, 69120 Heidelberg, Germany. [4] Department of Immunology, Weizmann Institute of Science, 234 Herzl Street, 76100 Rehovot, Israel. [5] Bernhard Nocht Institute for Tropical Medicine, Bernhard-Nocht-Straße 74, 20359 Hamburg, Germany. [6] Division of Developmental Immunology, German Cancer Research Center (DKFZ), Im Neuenheimer Feld 280, 69120 Heidelberg, Germany. [7] Division of Epigenomics and Cancer Risk Factors, German Cancer Research Center (DKFZ), Im Neuenheimer Feld 280, 69120 Heidelberg, Germany. [8] Division of Applied Bioinformatics, German Cancer Research Center (DKFZ), Im Neuenheimer Feld 280, 69120 Heidelberg, Germany. [9] Genomics and Proteomics Core Facility, German Cancer Research Center (DKFZ), Im Neuenheimer Feld 280, 69120 Heidelberg, Germany. [10] Division of Biostatistics, German Cancer Research Center (DKFZ), Im Neuenheimer Feld 280, 69120 Heidelberg, Germany. [11] Division of Epigenetics, German Cancer Research Center (DKFZ), Im Neuenheimer Feld 280, 69120 Heidelberg, Germany. [12] Division of Cellular and Molecular Pathology, German Cancer Research Center (DKFZ), Im Neuenheimer Feld 280, 69120 Heidelberg, Germany. [13] Department of Internal Medicine, Technical University of Munich, Ismaninger Straße 22, 81675 Munich, Germany. [14] Department of Internal Medicine III, Hematology and Oncology, University Hospital Regensburg, Franz-Josef-Strauss-Allee 11, 93053 Regensburg, Germany. Correspondence and requests for materials should be addressed to M.F. (email: markus.feuerer@ukr.de)

Regulatory T cells (T$_{reg}$) are important mediators of peripheral tolerance, and their absence leads to catastrophic autoimmunity in men (IPEX[1]) and mice (Scurfy[2]). T$_{reg}$ cells are characterized by both expression of the hallmark transcription regulator Foxp3[3–5] and a unique epigenetic profile[6–9]. T$_{reg}$ cells specialize to fulfill their diverse regulatory functions[10]. They can engage defined molecular pathways to specifically suppress either T$_H$1-polarized, T$_H$2-polarized, or T$_H$17-polarized immune effector cells[11]. For instance, under T$_H$1 conditions, T$_{reg}$ cells up-regulate expression of the T$_H$1-specific transcription factor T-box 21 (T-bet) and accumulate at inflammatory sites[12]. Correspondingly, under T$_H$2 conditions, T$_{reg}$ cells express Gata-binding protein 3 (Gata-3) and interferon regulatory factor 4 (Irf4), and T$_{reg}$-specific IRF4-deletion leads to IL-4 cytokine production of effector T cells and lymphoproliferative disease[13]. Up-regulation of signal transducer of activated T cells 3 (Stat3) is critical for the capacity of T$_{reg}$ cells to control T$_H$17-mediated inflammation, while its T$_{reg}$-specific deletion results in enhanced IL-17 production by effector cells and intestinal inflammation[14]. Therefore, T$_{reg}$ cells integrate unique parts of T$_H$ subtype-specific transcriptional programs to specifically control the respective T$_H$-polarized immune response.

Recombination signal-binding protein for immunoglobulin kappa J region (Rbpj) is a transcription factor commonly known for its function as a co-factor during Notch signaling, translating extracellular signals into gene expression changes[15]. In the context of T cell differentiation and function, Rbpj has been associated with T$_H$1/T$_H$2 cell fate decisions[16,17]. Indeed, in CD4$^+$Foxp3$^-$ conventional T (T$_{conv}$) cells, Rbpj in a complex with the Notch intracellular domain (NICD) was shown to be critical for regulation of Gata-3, an important molecular switch for optimal T$_H$2 responses[18,19]. In contrast to this, forced expression of the NICD in T$_{reg}$ cells rendered them incapable of suppressing T effector cells and caused autoimmunity[20]. This indicates that, based on the cellular context, Rbpj and Notch have a different impact on cellular responses. While the importance of Rbpj is well documented in T$_H$2 subset polarization, its function in T$_{reg}$ cells remains unclear.

Here we unveil a previously unappreciated role of Rbpj in regulating the capacity of T$_{reg}$ cells to restrain T$_H$2 responses. Loss of Rbpj renders T$_{reg}$ cells more sensitive to T$_H$2-inducing conditions and fosters the extensive generation of Gata-3-positive tissue-type T$_{reg}$ cells.

## Results

**Deletion of *Rbpj* causes defined organ pathology.** We specifically deleted *Rbpj* in T$_{reg}$ cells by crossing *Foxp3*$^{Cre,YFP}$ mice with mice harboring floxed *Rbpj* alleles (called Δ/Δ). We compared these to littermate control *Foxp3*$^{Cre,YFP}$ mice with wildtype *Rbpj* alleles (termed WT). We closely monitored our mice for 20 weeks, and about 40% of mice spontaneously developed splenomegaly and lymphadenopathy within this time interval, while about 60% of animals remained healthy (Fig. 1a, b). We confirmed the T$_{reg}$-specific deletion of *Rbpj* on DNA, RNA, and protein level (Supplementary Fig. 1a–d). First, we analyzed Δ/Δ and WT mice for the presence of CD4+CD25+Foxp3+ T$_{reg}$ cells in spleen and other tissues (Fig. 1c). We observed a strong increase in the fraction of T$_{reg}$ cells among CD4$^+$ T cells from about 12% in WT spleens to 28% in spleens from affected Δ/Δ animals. In absolute numbers, lymph nodes and spleen from Δ/Δ animals harbored about 10–20 times more T$_{reg}$ cells than their WT counterparts (Fig. 1c, right panel). This increase was not seen in the thymus, indicating normal thymic T$_{reg}$ cell output, or in mesenteric lymph nodes. Analysis of CD44 and L-selectin (CD62L) indicated activation of the T$_{reg}$ compartment in

affected Δ/Δ mice (Fig. 1d). Furthermore, affected Δ/Δ animals showed a higher density of Foxp3-positive T$_{reg}$ cells by immunohistology in spleen (Fig. 1e) and lymph nodes (Supplementary Fig. 1e). Affected animals developed noticeable skin pathology at snout, abdominal and tail regions, which served as useful biomarker to identify sick animals. Affected skin areas showed thickening of the epidermis and mononuclear cell infiltrates in the corium (Fig. 1f). Global defects in T$_{reg}$ cell function normally lead to a severe autoimmune manifestation, with destructive immune cell infiltration in a diverse set of organs[2]. Detailed histological examination of different organs including small intestine, large intestine, stomach, kidneys, salivary gland, eye, liver, and lung showed no signs of obvious immune cell infiltration and tissue destruction (Supplementary Figure 2a, b), indicating that Rbpj deficiency in T$_{reg}$ cells did not lead to a global loss of T$_{reg}$-mediated immune control. This was supported by data from a standard in vitro suppression assay with TCR-stimulated T$_{responder}$ cells, were we did not detect significant changes in the in vitro suppressive potential (Supplementary Fig. 3). In summary, these data indicate that RBPJ deficiency affected a more specific segment of T$_{reg}$ function.

**Germinal center formation and B-cell polarization.** Given that the T$_{reg}$-specific deletion of *Rbpj* affected secondary lymphoid organs and skin, we performed gene expression analysis of total LN RNA from WT and affected Δ/Δ animals (Fig. 2a). The most strongly up-regulated genes among the 4388 differentially expressed probes were involved in immune globulin (Ig) chain rearrangement and antibody production (Fig. 2a, highlighted in blue). Furthermore, B cell-specific markers such as *Cd22* and *Cd19* were over-expressed in LN from Δ/Δ mice. In addition, *Il4* was increased and pointed towards T$_H$2 subtype polarization (Fig. 2a and Supplementary Fig. 4a). B cells also down-modulated CD62L, and total numbers of B cells expanded about 10–20-fold in affected animals (Fig. 2b). To determine whether this increase in absolute B cell numbers and their maturation lead to changes in antibody Ig subtype distribution, we analyzed blood serum Ig levels via ELISA (Fig. 2c). Interestingly, we detected a significant up-regulation of IgG1 and IgE, while IgG3 was repressed in the serum (Fig. 2c), indicative of a classical IL-4 (=T$_H$2)-induced B cell antibody class switch[21]. As another hallmark of ongoing B cell differentiation and antibody production, we detected the formation of numerous germinal centers in LNs from affected Δ/Δ animals, but not in healthy WT animals (Fig. 2d and Supplementary Fig. 4b). Since IgE levels showed an almost 100-fold increase in affected Δ/Δ animals (Fig. 2c), and skin-resident mast cells express a high-affinity IgE receptor, we performed Giemsa staining of affected skin tissue. Indeed, we found an accumulation of mast cells rich in dark-stained granulae, presumably contributing to skin pathology in Δ/Δ animals (Fig. 2e and Supplementary Fig. 4c).

Because of the spontaneous germinal center formation, we analyzed whether the produced antibodies could bind self-antigens. To this end, protein was extracted from different organs isolated from *Rag2*-deficient animals, separated by SDS–PAGE, and incubated with serum from WT or affected Δ/Δ animals. We observed antibody binding to self-proteins extracted from organs such as lung, stomach, small intestine, pancreas, and eye with blood serum from Δ/Δ animals, although antibody-binding patterns were different between individual animals (Supplementary Fig. 5). In conclusion, our B-cell analysis revealed spontaneous germinal center formation with T$_H$2-specific Ig class-switch.

**T$_H$2 polarization of T$_{conv}$ cells.** *Il4* expression was increased in LN from Δ/Δ mice (Fig. 2a and Supplementary Fig. 4a), therefore

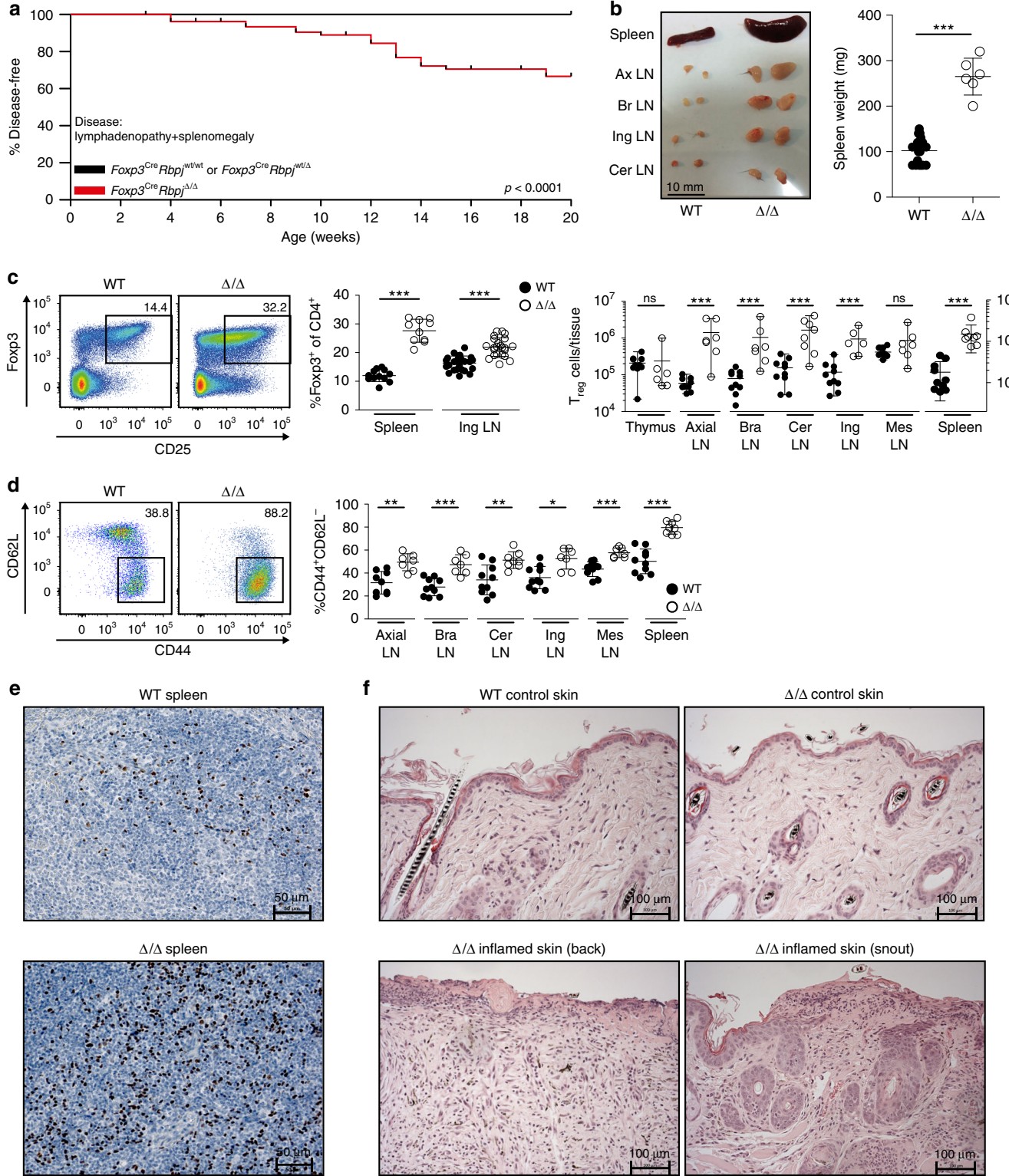

we studied $T_{conv}$ polarization. In spleen and LN, $T_{conv}$ cells became activated and differentiated into effector/memory T cells by down-regulation of CD62L and up-regulation of CD44. Furthermore, the absolute number increased by about 5–10-fold (Fig. 3a). Treatment of T cells from WT and affected Δ/Δ animals with phorbol-12-myristate-13-acetate (PMA) and Ionomycin to measure intracellular cytokine expression revealed higher frequencies of $T_{conv}$ cells from affected Δ/Δ animals producing IL-2, IL-4, and IL-13, while $T_{reg}$ cells remained unchanged (Fig. 3b).

This indicated a $T_H2$ polarization. Since Gata-3 is the master transcription factor of $T_H2$ cells, we analyzed Gata-3 expression in the $T_{conv}$ compartment and detected increased frequencies of Gata-3-positive $T_{conv}$ cells from about 3% in WT to 13% in Δ/Δ animals by flow cytometry (Fig. 3c, d), which was confirmed on RNA level (Fig. 3d, right panel). Co-staining of stimulated $T_{conv}$ cells with IL-4, IL-13, and Gata-3 revealed that IL-4 and IL-13 was mainly produced by Gata-3-positive $T_H2$ cells (Fig. 3c). To examine closely the link between the observed $T_H2$ bias and the

**Fig. 1** $T_{reg}$-specific deletion of *Rbpj* causes lymphoproliferative disease. **a** Kaplan–Meier survival curve illustrating disease development in *Foxp3*[Cre]*Rbpj*[Δ/Δ], *Foxp3*[Cre]*Rbpj*[wt/Δ], and *Foxp3*[Cre]*Rbpj*[wt/wt] animals within 20 weeks after birth. Disease defined by inflammatory skin lesions, lymphadenopathy, and splenomegaly. We observed 79 animals with $T_{reg}$ lineage-specific bi-allelic *Rbpj* deletion (*Foxp3*[Cre]*Rbpj*[Δ/Δ]), 67 animals with mono-allelic *Rbpj* deletion (*Foxp3*[Cre]*Rbpj*[wt/Δ]), and 79 wildtype animals (*Foxp3*[Cre]*Rbpj*[wt/wt]). Statistical testing log-rank Mantel–Cox test ($p < 0.0001$). $T_{reg}$-specificity of *Rbpj* deletion in Supplementary Fig. 1. **b** Splenomegaly and lymphadenopathy in affected Δ/Δ animals, representative picture (Ax: axial; Br: brachial; Ing: inguinal; Cer: cervical). Right panel, spleen weight in milligram ($n = 6$–19, Mann–Whitney test) in a dot plot where error bars indicate standard deviation and center line mean value. **c** Quantification of $T_{reg}$ (CD3[+]CD4[+]CD8[−]CD25[+]Foxp3[+]) cell number and frequency in WT vs. affected Δ/Δ animals. Left panel, representative pseudocolor plots of splenic $T_{reg}$ cells of CD4[+] T cells, frequencies shown as number. Middle graph, $T_{reg}$ frequencies in spleen and LN (% of CD4[+], $n = 6$–26, Mann–Whitney test). Black dots $T_{reg}$ cells from WT, open circle dots $T_{reg}$ cells from Δ/Δ animals, dots represent individual mice, line mean value. Right graph, total $T_{reg}$ cell numbers in various tissues ($n = 6$–10, Mann–Whitney test, Mes: mesenteric). **d** Left panel, CD44 and CD62L expression in $T_{reg}$ cells from spleens of WT and affected Δ/Δ animals, quantification for several tissues to the right ($n = 7$–10, Mann–Whitney test). **e** Immunohistochemistry (IHC) of spleen from WT and affected Δ/Δ animals. Foxp3 staining in brown with hematoxylin staining. **f** Hematoxylin and eosin (H&E) staining of non-inflamed and inflamed skin tissue of representative WT and affected Δ/Δ mouse. Additional stainings in Supplementary Fig. 2. Data representative of two or more independent experiments with individual mice. Asterisks indicate statistical significance with ***$p < 0.001$, **$p < 0.01$, and *$p < 0.05$. In IHC and H&E stainings, original magnification scale bars have been magnified for better visibility. Source data are provided as a Source Data file

profound B cell phenotype, we analyzed T-follicular helper (Tfh) cells in LNs by Cxcr-5 and PD-1 co-staining. We observed an increase in the Cxcr-5[+]PD-1[+] Tfh cell fraction and total numbers (Fig. 3e). Furthermore, we identified significantly more Gata-3-polarized Tfh cells in Δ/Δ animals (Fig. 3f).

**Environmental conditions influence disease development.** Approximately 40% of *Foxp3*[Cre,YFP]*Rbpj*[Δ/Δ] mice developed a $T_H2$-polarized disease, while 60% of animals did not get sick during a 20-week observation period (Fig. 1a). Since most mice that developed a pathology were older than 10 weeks, environment-related causes might influence the phenotype in Δ/Δ animals. Therefore, we transferred embryos into a new breeding facility with individually ventilated cages and a defined altered Schaedler flora (details in Methods section). In this new environment, the incidence rate dropped and about 90% of animals remained healthy until 20 weeks of age (Fig. 4a). These findings indicate that environmental factors influence $T_H2$-disease onset in Δ/Δ animals. To further investigate this, we infected healthy young Δ/Δ and control animals from the new breeding facility with the parasitic nematode *Strongyloides ratti* (*S. ratti*). Infective larvae penetrate the skin and migrate within 3 days to the small intestine. Normally, immune competent mice mount a canonical $T_H2$ response and resolve the infection within 2–4 weeks[22]. Final expulsion of parasites from the intestine is predominantly mediated by IL-9 and activated mucosal mast cells[23]. Especially, Foxp3[+] $T_{reg}$ cells have been shown to expand following *S. ratti* infection[24]. Therefore, we infected two cohorts of Δ/Δ and WT animals with *S. ratti* infectious larvae into the footpad and examined animals after 6 (cohort 1) and 14 days (cohort 2, Fig. 4b). Already 6 days post infection (p.i.), $T_{reg}$ frequency of CD4[+] T cells in the spleen increased in Δ/Δ animals to about 28%, while WT animals had normal $T_{reg}$ frequencies at that time point (about 9%) (Fig. 4c). The three-fold increase in $T_{reg}$ frequency was persistent and also measured at 14 days p.i. (Fig. 4c). At the 14-day time point, $T_{conv}$ cells expressed more Gata-3 in Δ/Δ animals (Fig. 4d), as well as more IL-4, IL-13, and IL-9, but not IFN-γ (Fig. 4e–g). The fraction of Gata-3-high expressing $T_{reg}$ cells increased in the Δ/Δ animals from day 6 to day 14 p.i. to 56%, while at the same 14-day time point, the WT animals had significant lower numbers (27%, Fig. 4d). Furthermore, we detected increased levels of IgE, but not IgM, in blood serum of Δ/Δ animals (Fig. 4h), mirroring the phenotype observed during our initial breeding (Fig. 2c). Interestingly, on day 6, Δ/Δ animals with stronger $T_H2$-polarization and increased IL-9 production displayed reciprocally reduced output of *S. ratti* DNA (Fig. 4i) and a strong trend towards reduced numbers of

parasitic females in the intestine (Fig. 4j). In summary, our data indicate that disease development in Δ/Δ animals was influenced by environmental factors, such as the breeding environment, and that the $T_H2$-inducing parasite *S. ratti* could trigger the onset of this phenotype in Δ/Δ animals.

**Characterization of Rbpj-deficient $T_{reg}$ cells.** Mice with Rbpj-deficiency in $T_{reg}$ cells spontaneously developed a $T_H2$-type disease. Since the Rbpj-deficiency was specific to $T_{reg}$ cells, we aimed at dissecting the molecular properties of those. Therefore, we measured the expression of classical $T_{reg}$ associated proteins, such as Foxp3, cytotoxic T-lymphocyte-associated protein 4 (Ctla-4), or Ikaros family zinc finger 2 (Helios), and did not detect obvious differences between $T_{reg}$ cells from WT or affected Δ/Δ animals (Fig. 5a). Analogously, the methylation status of the $T_{reg}$-specific demethylated region (TSDR), a well-described methylation-sensitive *cis*-regulatory region required for durable expression of the *Foxp3* gene[25], was unchanged (Fig. 5b). The investigation of the T cell receptor (TCR) repertoire revealed no abnormal TCR beta J-chain usage (Fig. 5c), nor any dominant clones (Fig. 5d), and only a small increase in overall clonality and decreased entropy (Fig. 5e). As $T_{reg}$ cells were abundantly present in lymphatic organs of affected Δ/Δ animals, we analyzed peripheral tissues. $T_{reg}$ frequency and total numbers were either equal or elevated (40-fold in affected skin tissue and about three-fold in the lung, Fig. 5f–g and Supplementary Fig. 6) in affected Δ/Δ animals. Since $T_{reg}$ cells co-opt parts of $T_H$ subset programs to specifically suppress those $T_H$ subset responses, we analyzed Gata-3 protein expression. Indeed, we could detect a strong increase of Gata-3 high-expressing $T_{reg}$ cells isolated from affected Δ/Δ compared to WT animals (about 45% vs. 12%, respectively), which could be confirmed on RNA level (Fig. 5h). Furthermore, we investigated T follicular regulatory cells based on the expression of Cxcr5[+] and PD1[+] (Tfr; CD4[+]CD25[+]Foxp3[+]Cxcr5[+]PD1[+]) in WT and Δ/Δ animals. The total number of PD1[+] Tfr cells was increased in LNs of affected Δ/Δ animals (Fig. 5i), excluding the possibility that a loss of Tfr cells was responsible for the observed phenotype. In addition, we co-stained PD-1 and Gata-3 and noticed a strong increase in Gata-3[+]PD-1[+] $T_{reg}$ cells in Δ/Δ mice (Fig. 5j).

**Altered gene expression in Rbpj-deficient $T_{reg}$ cells.** To dissect molecular characteristics, we performed array-based gene expression analysis of $T_{reg}$ cells isolated from affected Δ/Δ and WT animals (Fig. 6a). Six hundred and ninety probes were found to be differentially expressed. For example, we observed a strong up-regulation of IL-7 receptor (*Il7r*) and Killer cell lectin-like receptor subfamily G member 1 (*Klrg1*) genes, while Bcl-2-like

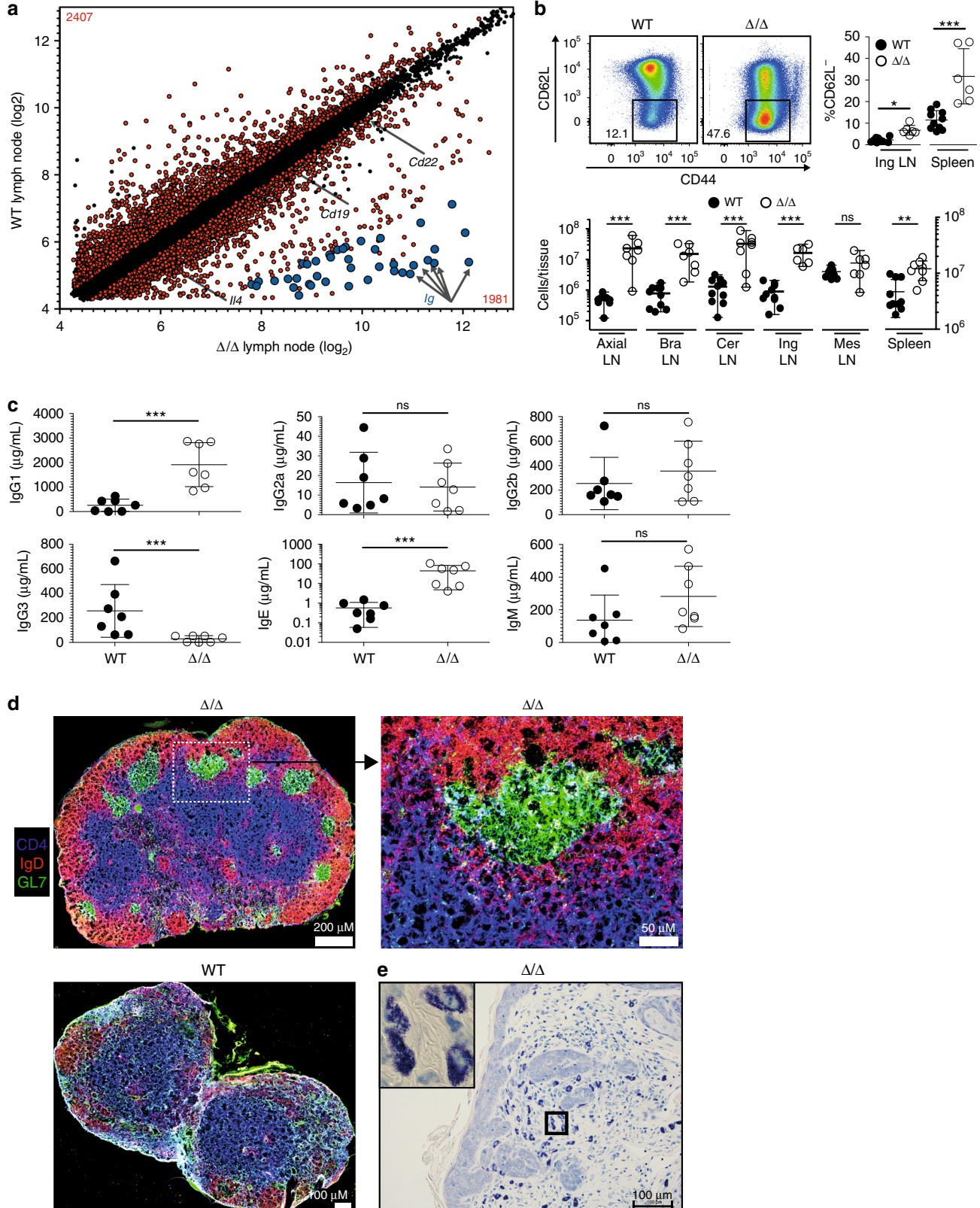

protein 11 (*Bcl2l11*) and Deltex-1 (*Dtx1*) were under-represented in Δ/Δ T$_{reg}$ cells (Fig. 6a, b). *Bcl2l11* serves as translocator of apoptosis-inducing factors and regulator of mitochondrial depolarization[26,27]. Indeed, Rbpj-deficient T$_{reg}$ cells showed less caspase-3 activity when compared to WT T$_{reg}$ cells (Fig. 6c and Supplementary Fig. 7). Besides differences in apoptosis, Rbpj-

deficient T$_{reg}$ cells also up-regulated the *Il7r* gene encoding for the IL-7 receptor (IL-7R, CD127). In thymus and spleen, about 75% of T$_{conv}$ cells expressed the IL-7R, and this was not different in WT and Δ/Δ animals. In contrast to this, T$_{reg}$ cells from Δ/Δ mice showed a strong increase in IL-7R expression in the spleen (Fig. 6d). Since the IL-7R is involved in survival and

**Fig. 2** Analysis of B cell involvement. **a** Gene expression profile of LNs from WT vs. affected $\Delta/\Delta$ animals. Significantly different probes colored in red and numerated in corners ($p < 0.05$), three biological replicates per group. Of 50 most up-regulated probes in $\Delta/\Delta$ LNs, 38 probes annotated Ig-related genes (highlighted blue). Statistical testing described in Methods section. For few genes, differential expression confirmed in Supplementary Fig. 4. **b** Upper left panel, representative dot plots of CD62L and CD44 expression of splenic B cells (CD3⁻CD4⁻CD8⁻CD19⁺) from WT vs. affected $\Delta/\Delta$ animals, quantification for spleen and LN B cells to the right ($n = 6$–10, Mann–Whitney test). Lower panel absolute B cell numbers per lymph node or total spleen ($n = 6$–10, Mann–Whitney test). Black dots B cells in WT animals, open circles $\Delta/\Delta$-mice, individual mice are shown. **c** Ig subtype analysis of antibodies in peripheral blood serum of WT vs. affected $\Delta/\Delta$ animals. Ig subtype levels detected by ELISA. Statistical testing with Mann–Whitney test ($n = 7$). **d** Immunohistochemistry of lymph nodes from WT and affected $\Delta/\Delta$ animals. CD4 staining in blue, IgD staining in red, and GL7 staining in green. Top left image scanned lymph node from $\Delta/\Delta$ animal. Typical germinal center highlighted with box and enlarged in image to the right. At bottom left, image of WT lymph node. All images recorded with same settings and color intensity adjustments. Additional wildtype stainings in Supplementary Fig. 4. **e** Giemsa staining of skin tissue from affected $\Delta/\Delta$ animal. Granule-rich mast cells stained in blue. Mast cell enriched area outlined by rectangle, enlarged on the upper left quadrant of image. Additional wildtype stainings in Supplementary Fig. 4. Data are representative of two or more independent experiments with individual mice (**b**) or a single experiment with individual mice (**a**, **c**, **d**, **e**). In IHC and Giemsa stainings, original magnification scale bars magnified for better visibility. Source data are provided as a Source Data file

proliferation[28], we correlated IL-7R expression with the overall frequency of $T_{reg}$ cells in unaffected, as well as phenotypically affected $\Delta/\Delta$ animals of varying age. A good correlation between both parameters could be observed ($r = 0.81$), suggesting that IL-7R expression supports $T_{reg}$ accumulation (Fig. 6e). To further validate this, we co-stained IL-7R expression with KI-67, a widely accepted cell proliferation marker[29] (Fig. 6f). The majority of $T_{reg}$ cells in $\Delta/\Delta$ animals expressed both IL-7R and KI-67, which was not seen in WT animals (Fig. 6f), indicating that IL-7R high-expressing $T_{reg}$ cells were the proliferating fraction in $\Delta/\Delta$ animals. Next, we measured the phosphorylation of Stat-5, a downstream component of the IL-7R signaling cascade[30]. With escalating doses of IL-7, Rbpj-deficient $T_{reg}$ cells phosphorylated significantly more Stat-5, indicating elevated cytokine sensitivity (Fig. 6g). Since Klrg1 was up-regulated in the gene expression profile (Fig. 6a), we co-stained Klrg1 and IL-7R expression (Fig. 6h). Indeed, while a Klrg1⁺IL7R⁺ double-positive population was almost absent in $T_{conv}$ cells or WT $T_{reg}$ cells, the spleens of affected $\Delta/\Delta$ animals harbored about 60% Klrg1⁺IL7R⁺ $T_{reg}$ cells (Fig. 6h). Therefore, both IL-7R and Klrg1 were valuable parameters to identify the intensively proliferating subpopulation of $T_{reg}$ cells in affected $\Delta/\Delta$ mice.

**IL-7R⁺Klrg1⁺ $T_{reg}$ cells are reminiscent of tisT$_{reg}$ST2.** In a recent publication, we described a tissue-resident $T_{reg}$ population characterized by the expression of, amongst others, IL-7R, Klrg-1, IL-33 receptor alpha (ST2), and Gata-3[7]. This $T_{reg}$ subset, mainly present within tissues and $T_H$2-polarized, was called tisT$_{reg}$ST2[7]. Since we detected a stong enrichment of Gata-3, Klrg1, and IL-7R-expressing $T_{reg}$ cells specifically in affected $\Delta/\Delta$ animals, we performed a co-staining for ST2 and Klrg1 (Fig. 7a). About 2% of $T_{reg}$ cells from spleens of WT animals co-express ST2 and Klrg1 compared to 50–60% of spleen $T_{reg}$ cells from affected $\Delta/\Delta$ animals (Fig. 7a). A co-staining with KI-67 and Gata-3 revealed that Klrg1⁺ST2⁺ $T_{reg}$ cells in both WT and $\Delta/\Delta$ animals were $T_H$2 polarized and proliferating (Fig. 7a, middle panel). Klrg1⁺ST2⁺ were also significantly increased in lymph nodes (WT: 2%, $\Delta/\Delta$: 27%) and skin tissue (WT: 48%, $\Delta/\Delta$: 65%, Fig. 7b). To compare the tissue-like gene expression program of Klrg1⁺ tisT$_{reg}$ST2-like cells in $\Delta/\Delta$ animals on a broader scale, we sorted $\Delta/\Delta$ Klrg1⁺, $\Delta/\Delta$ Klrg1⁻, WT Klrg1⁺ and WT Klrg1⁻ $T_{reg}$ cells from spleen and performed RNA sequencing (RNA-seq) analysis. We extracted RNA-seq data from fat, skin and LN-derived bulk $T_{reg}$ cells from a previous study[7] and normalized all datasets. We then plotted 106 reported tisT$_{reg}$ST2 genes in a heatmap (Fig. 7c). Interestingly, there was a strong gene expression overlap between fat and skin $T_{reg}$-differentially regulated genes with spleen $\Delta/\Delta$-derived and WT-derived Klrg1⁺ $T_{reg}$ cells, indicating that the majority of $\Delta/\Delta$-$T_{reg}$ cells from affected mice indeed displayed a

tisT$_{reg}$ST2-like signature. Still, when listing key genes identifying tissue $T_{reg}$ cells from fat (*Pparg*) or skin (*Gpr55*), as well as tissue $T_{reg}$ effector molecules such as *Il10* and amphiregulin (*Areg*), the $\Delta/\Delta$ Klrg1⁺ tisT$_{reg}$ST2-like population in the spleen of $\Delta/\Delta$ animals did not express comparable levels of these markers (Fig. 7d). This indicated that they were generated in the lymphoid tissue rather than extravasated from non-lymphoid tissues. To analyze differences in more detail, we prepared MA plots for comparisons between all four groups (Fig. 7e). The comparison between WT Klrg1⁺ vs. $\Delta/\Delta$ Klrg1⁺ $T_{reg}$ cells revealed 2036 differential expressed genes, which include the molecular changes associated with loss of Rbpj (Fig. 7e; left panel). This group contains well-known Rbpj target, such as *Dtx1*[31,32], as well as suppression-related proteins such as *Id-3*[33]. The comparison between WT Klrg1⁻ vs. $\Delta/\Delta$ Klrg1⁺ $T_{reg}$ cells revealed 3330 differential expressed genes (Fig. 7e; right panel), which include tissue-$T_{reg}$-related genes, such as *Klrg1* and *Il1rl1*, as well as suppression-related proteins such as *Bach2*[34]. In summary, we showed that affected $\Delta/\Delta$ animals harbor a strongly increased tisT$_{reg}$ST2-like population in their lymphoid tissues.

**Genome-wide chromatin accessibility of WT and $\Delta/\Delta$ $T_{reg}$ cells.** In affected $\Delta/\Delta$ animals, Klrg1⁺ tissue-like $T_{reg}$ cells constitute the majority of all spleen $T_{reg}$ cells, while in WT animals, Klrg1⁻ non-tissue $T_{reg}$ cells dominate the $T_{reg}$ pool at large. To obtain insights into their gene-regulatory landscapes, we isolated both populations and performed the Assay for Transposase-Accessible Chromatin using sequencing (ATAC-seq)[35]. In total, across cell types and replicate experiments, we detected 68,214 ATAC-seq peaks throughout the genome (Fig. 8a). Reference genome annotation revealed that 17.5% of the peaks located to promoters, 41.5% to introns, 2.0% to exons, and 39% to intergenic genomic regions (Fig. 8b). In $\Delta/\Delta$ $T_{reg}$ cells, about 3400 regions were more accessible compared to WT $T_{reg}$ cells, while in WT $T_{reg}$ cells, 10,816 regions were more accessible in comparison to $\Delta/\Delta$ $T_{reg}$ cells (Fig. 8b, c). We next asked which transcription factor motifs were enriched in differential accessible chromatin regions to identify potential drivers of WT-specific and $\Delta/\Delta$ $T_{reg}$-specific gene-regulatory programs. De novo motif analysis revealed a strong Gata transcription factor signature in $\Delta/\Delta$ $T_{reg}$-specific regions (24.38%) vs. background sequences (13.22%) with a high $p$-value ($10^{-68}$) and score (0.95) (Fig. 8d, Supplementary Fig. 8a). Of the relevant Gata family members, only Gata-3 was significantly induced in $\Delta/\Delta$ $T_{reg}$ cells (Fig. 8d, lower panel). In addition, we identified enrichment of Ets, Klf, and AP-1-binding sites in $\Delta/\Delta$ $T_{reg}$-specific regions (Supplementary Fig. 8a). WT $T_{reg}$-specific regions were dominated by Ets, Tcf, Stat, as well as Nur77 motifs, and we also identified significant enrichment of a motif highly similar to the recently described Rbpj consensus-

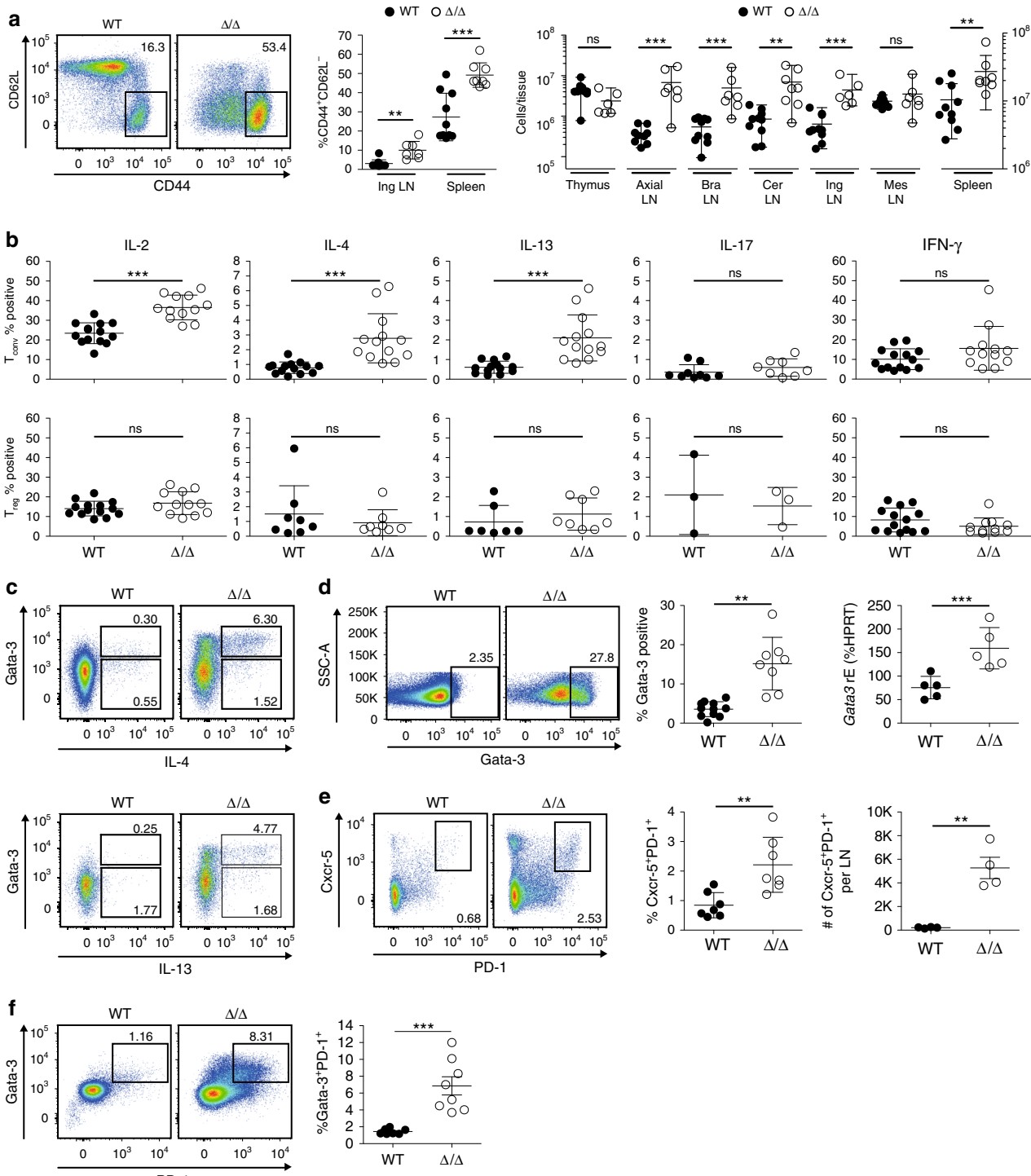

**Fig. 3** $T_{conv}$ cells derived from affected $\Delta/\Delta$ animals are $T_H2$ polarized. **a** $T_{conv}$ cells (CD3+CD4+CD25−Foxp3−) from spleens of WT and affected $\Delta/\Delta$ animals stained for CD44 and CD62L expression, quantification for spleen and LN ($n = 7$–10, Mann–Whitney test) adjacent to pseudocolor plots. Right panel, $T_{conv}$ cell numbers per tissue ($n = 6$–10, Mann–Whitney test). Black dots $T_{conv}$ cells in WT animals, open circles $\Delta/\Delta$-derived $T_{conv}$ cells, individual mice are shown. **b** Restimulation of splenic T cells followed by intracellular cytokine staining. Upper panel intracellular cytokines in $T_{conv}$ cells, lower panel in $T_{reg}$ cells (CD3+CD4+CD25+Foxp3+). Statistical testing with Mann–Whitney test (IL-2: $n = 12$–14; IL-4: $n = 8$–14; IL-13: $n = 7$–13; IL-17: $n = 3$–9; IFN-γ: $n = 12$–14). **c** Co-staining of stimulated $T_{conv}$ cells with cytokine and Gata-3 antibody. Upper panel representative dot plots of PMA/Ionomycin-stimulated WT (left) or $\Delta/\Delta$ (right) $T_{conv}$ cells, with IL-4 staining on X-axis and Gata-3 staining on Y-axis. Lower panel, IL-13 staining. **d** Left, dot plots of Gata-3 protein staining in spleen-derived $T_{conv}$ cells; middle panel, quantification of Gata-3-positive cells (% of $T_{conv}$, $n = 8$–11, unpaired t-test); right panel, *Gata3* mRNA in $T_{conv}$ cells (% *Hprt*, $n = 5$, unpaired t-test). **e** Identification of T follicular helper cells in lymph nodes from WT vs. affected $\Delta/\Delta$ animals. Tfh cells identified as CD3+CD4+CD8−CD25−Foxp3-Cxcr-5+PD-1+ T cells and quantified ($n = 7$, unpaired t-test). Total numbers of Tfh cells per LN to the right ($n = 4$, unpaired t-test). **f** CD3+CD4+CD8−CD25−Foxp3− T cells in lymph nodes of WT or affected $\Delta/\Delta$ animals co-stained with PD-1 and Gata-3. Quantification to the right ($n = 8$, unpaired t-test). Data are representative of two or more independent experiments with individual mice (**a**, **b**, **d**, **e**, **f**) or a single experiment with individual mice (**c**). Source data are provided as a Source Data file

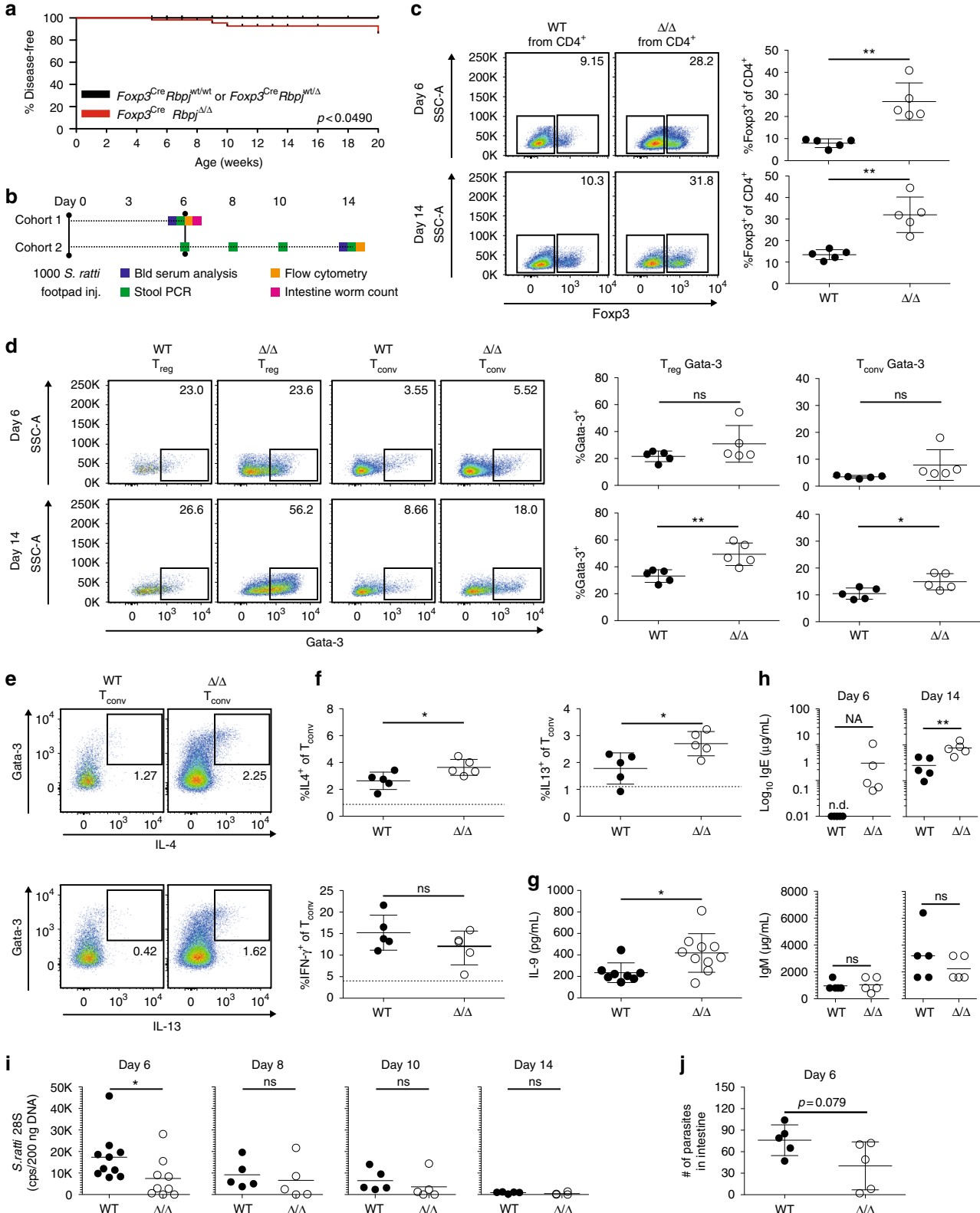

binding site (Supplementary Fig. 8b). Examples of ATAC-seq signals along with occurrences of Rbpj motifs (both de novo identified and previously identified consensus motifs) at key genes are shown in Fig. 8e–h and Supplementary Fig. 8c. Earlier, we showed that Foxp3 expression was unchanged between Δ/Δ and WT $T_{reg}$ cells (Fig. 5a) and Foxp3 TSDR demethylation was unaffected by loss of Rbpj (Fig. 5b). Accordingly, the ATAC-seq

profile at the Foxp3 locus remained unchanged between both groups (Fig. 8e). Peak calling identified the TSDR region as highly accessible region in both WT and Δ/Δ $T_{reg}$ cells, but no Rbpj-binding motif has been detected in this or any other part of the Foxp3 gene. The Rbpj locus was accessible in both WT and Δ/Δ $T_{reg}$ cells, indicating that Rbpj was not required to open its own locus (Supplementary Fig. 8c). In contrast to this, tissue-$T_{reg}$-

**Fig. 4** Disease development in Δ/Δ animals is environment-related. **a** Kaplan–Meier curve illustrating disease development in WT vs. Δ/Δ animals within 20 weeks after birth as in Fig. 1a, but in different breeding environment (55 animals Foxp3[Cre]Rbpj[Δ/Δ], 45 animals Foxp3[Cre]Rbpj[wt/Δ], 40 animals Foxp3[Cre]Rbpj[wt/wt]). Statistical testing log-rank Mantel–Cox test ($p = 0.049$). **b** Parasitic infection regimen with S. ratti. Two cohorts, each 5 WT vs. 5 healthy young Δ/Δ animals, infected on day 0. End point for cohort 1, 6 days and cohort 2, 14 days post infection (p.i.). Experiments and parameter measurement intervals indicated. **c** Representative pseudocolor plots displaying Foxp3 expression in WT vs. Δ/Δ spleen CD4[+] T cells 6 or 14 days p.i.. Right, quantification of $T_{reg}$ (CD3[+]CD4[+]Foxp3[+]) cell percentage of CD4[+]. Black dots WT, open circle Δ/Δ animals ($n = 5$, unpaired $t$-test). **d** Gata-3 staining in $T_{reg}$ and $T_{conv}$ (CD3[+]CD4[+]Foxp3[−]) cells from WT vs. Δ/Δ animals infected with S. ratti 6 or 14 days p.i., quantification right (%Gata-3[+] of $T_{reg}$ or $T_{conv}$, $n = 5$, unpaired $t$-test). **e** Co-staining of stimulated $T_{conv}$ cells isolated from WT or Δ/Δ animals 14 days p.i. with IL-4 and IL-13 and Gata-3 antibody. **f** Quantification of intracellular cytokine expression in re-stimulated splenic $T_{conv}$ cells ($n = 5$, unpaired $t$-test). Frequencies of IL-4, IL-13, and IFN-γ. Dotted line unstimulated, but transport inhibitor-treated levels. **g** Mesenteric LN cells of WT vs. Δ/Δ animals 6 and 14 days p.i. stimulated with anti-CD3 and detection of secreted IL-9 by ELISA ($n = 10$, unpaired $t$-test). **h** Ig subtype analysis of antibodies in peripheral blood serum of WT vs. Δ/Δ animals 6 or 14 days p.i. Ig subtype levels detected by ELISA ($n = 5$, paired $t$-test). **i** Copy numbers of S. ratti 28S DNA in 200 ng of mouse stool samples. Day 6, stool samples from both cohorts 1 and 2 ($n = 10$), on day 8, 10, and 14 only from cohort 2 ($n = 5$, unpaired $t$-test). **j** Total count of parasites in the intestine on day 6 (cohort 1, unpaired $t$-test). Data are representative of two independent experiments (cohorts) with several individual mice ($n = 5$). Source data are provided as a Source Data file

related genes, such as *Klrg1* or *Il1rl1* (ST2) showed significant ATAC-seq signals around the promoter and potential enhancer sites in Δ/Δ $T_{reg}$ cells, and Rbpj-binding motifs were also found in these regions (Fig. 8f). These changes in ATAC-seq signals translated into enhanced expression of *Il1rl1* and *Klrg1* (Fig. 8i). In addition to tissue $T_{reg}$-related genes, $T_{reg}$ suppressive function-associated genes, such as *Il2ra*, *Dtx1*, and *Bach2* also displayed a distinct ATAC-seq profile: in intragenic and/or enhancer sites, WT $T_{reg}$ cells had enriched signals compared to Rbpj-deficient $T_{reg}$ cells (Fig. 8g). Again, Rbpj-binding motifs were detected in differential peaks and ATAC-seq profiles correlated well with changes in gene expression: *Il2ra* gene expression was significantly down-modulated in Δ/Δ $T_{reg}$ cells, and expression of *Dtx1* and *Bach2* was almost completely lost (Fig. 8i). Interestingly, *Bach2* has recently been described as a key transcription factor involved in regulating $T_H2$ polarization by inhibiting Gata-3 expression[34]. Other $T_H2$-polarized-related regions were also differentially accessible in Δ/Δ $T_{reg}$ cells, e.g., the $T_H2$ locus control region (*Rad50*), *Il10* and *Areg*, but the corresponding genes were not expressed (Fig. 8h, Fig. 7d and Supplementary Fig. 8c). Taken together, our data suggest that both Rbpj and Gata3 influence the expression of key tis$T_{reg}$ST2-related genes, and that the genomic deletion of Rbpj in concert with a Gata-3-inducing $T_H2$-type inflammatory environment lead to the massive differentiation and expansion of tis$T_{reg}$ST2-like cells in Δ/Δ animals.

**$T_H2$-polarized $T_{reg}$ fail to suppress $T_H2$ responses in vitro.** But why are these $T_H2$-polarized $T_{reg}$ cells not controlling the $T_H2$-response anymore? Recently, it was shown that *Lilrb4a* (encoding the protein ILT3) expressing $T_{reg}$ cells were unable to regulate $T_H2$-responses due to their inability to control the maturation of a specific $T_H2$-promoting DC subset[36]. This subset of DCs is characterized by the expression of PD-L2 and IRF-4[36–38]. Our ATAC-seq data identified a highly accessible region at the *Lilrb4a* promoter in Klrg1[+]ST2[+] $T_{reg}$ cells isolated from affected Δ/Δ animals, and a Rbpj-binding site was also predicted in this region (Fig. 9a). Indeed, enhanced activity at the *Lilrb4a* promoter resulted in increased *Lilrb4a* expression in Klrg1[+]ST2[+] $T_{reg}$ cells from affected Δ/Δ animals (Fig. 9b). In addition, Klrg1[+]ST2[+] $T_{reg}$ cells from WT animals expressed more ILT-3 than Klrg1[−]ST2[−] non-tissue type $T_{reg}$ cells, suggesting a general mechanism of ILT3-expression during differentiation of the tis$T_{reg}$ST2-like gene expression program. Our de novo motif analysis indicated that Gata-3 was responsible for large parts of the tis$T_{reg}$ST2-like signature in Klrg1[+]ST2[+] $T_{reg}$ cells from affected Δ/Δ animals (Fig. 8d). To study the link between ILT3[+]Gata-3 overexpression and control of $T_H2$ responses, we performed in vitro polarization studies with $T_{reg}$ cells. IL-4 is the prototype cytokine to induce

Gata-3 expression and $T_H2$ differentiation, and IL-33 is linked to the generation of tis$T_{reg}$ST2 cells[7]. Therefore, we FACS-sorted highly pure $T_{reg}$ cells from *Foxp3*[GFP] animals and expanded them with anti-CD3/CD28 microbeads, IL-2, IL-4, and IL-33 or without the latter two cytokines as control, for 6 days. Both groups of expanded $T_{reg}$ cells stayed highly Foxp3 positive (Fig. 9c). Interestingly, we were able to co-induce ILT3 and Gata-3 expression specifically in the IL-4 and IL-33-treated $T_{reg}$ cells (Fig. 9c). Using this model, we studied the ability of ILT3-expressing $T_H2$-polarized $T_{reg}$ cells to influence DC maturation and DC-mediated $T_H2$ polarization of FACS-sorted CD4[+]Foxp3[-]CD62L[+] naive T cells in vitro. Our data revealed that ILT3-expressing $T_H2$-polarized $T_{reg}$ cells profoundly promoted the differentiation of PD-L2[+]IRF4[+] DCs, a subset described to support $T_H2$ polarization in vivo[36–38] (Fig. 9d). In addition, ILT3-expressing $T_{reg}$ cells were unable to suppress the $T_H2$ differentiation of IL-4 and anti-CD3-stimulated naive T cells into Gata-3-polarized effector T cells in the presence of DCs (Fig. 9e). These data strongly indicate that overexpression of Gata-3 and ILT3 renders $T_{reg}$ cells less able to suppress $T_H2$ responses in vitro. Finally, we investigated the sensitivity of Rbpj-deficient $T_{reg}$ cells to $T_H2$-inducing conditions. To this end, we FACS-sorted and expanded Rbpj-deficient $T_{reg}$ cells from healthy, young animals, with no pre-existing $T_H2$ polarization, and compared them to WT $T_{reg}$ cells. Both groups were treated with escalating doses of IL-4 in vitro (Fig. 9f). Indeed, Rbpj-deficient $T_{reg}$ cells were more sensitive to the $T_H2$-inducing IL-4 treatment, translating into enhanced Gata-3 protein and mRNA induction in Δ/Δ $T_{reg}$ cells (Fig. 9f). This elevated sensitivity towards Gata-3 induction could explain the profound expansion of Gata-3[+]Klrg1[+]ST2[+] $T_H2$-polarized $T_{reg}$ cells in affected Δ/Δ animals, with ameliorated $T_H2$-suppressive potential.

## Discussion

In this study, we identify a previously unrecognized role for Rbpj in $T_{reg}$ cell-mediated immune homeostasis. Upon $T_{reg}$-specific Rbpj deletion in *Foxp3*[Cre]*Rbpj*[Δ/Δ] animals, mice developed a lymphoproliferative disease with type-2 effector polarized B-cell and T-cell responses. Disease development was environment-related and could be induced by infection with the parasitic nematode S. ratti. The finding that disease development was environment-related could explain the discrepancy to a published study using mice with *Rbpj*[Δ/Δ] $T_{reg}$ cells, where the authors did not report the lymphoproliferative characteristic[20].

But what happened once the proper environmental trigger has been received? Based on our data, we would argue that deleting Rbpj confined the functional capacity of $T_{reg}$ cells in several ways. First, augmented proliferation potential: the down-modulation of

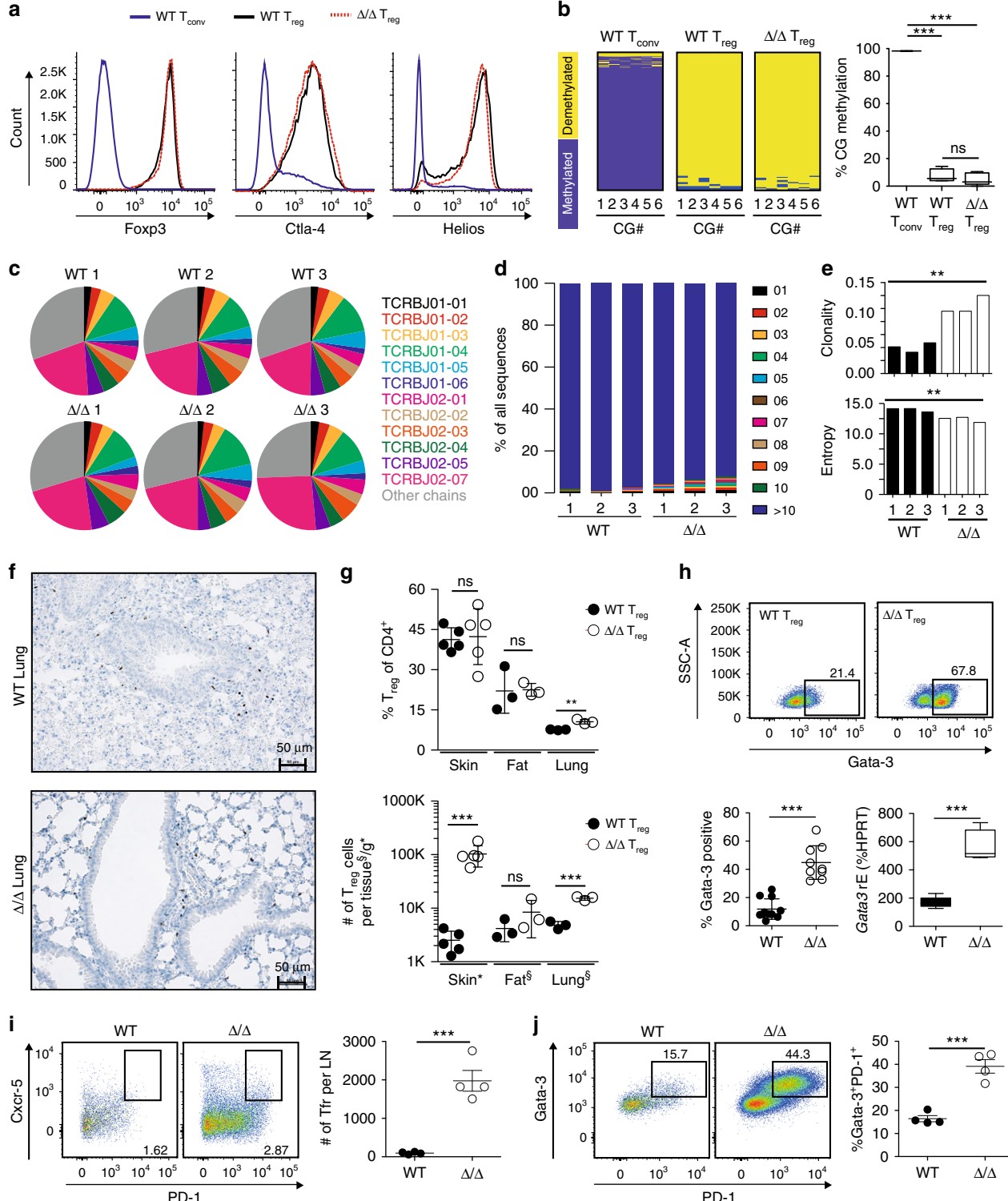

*Bcl2l11* could lead to enhanced resistance to apoptosis[26,27], while the up-regulation of the Interleukin-7 receptor promoted proliferation and supported a strong increase in total $T_{reg}$ numbers in lymphoid tissues. Potential Rbpj-binding sites at the *Il7r* promoter region have already been reported[39]. Second, Rbpj is important to restrict the $T_H2$ differentiation potential of $T_{reg}$ cells: Rbpj-deficient $T_{reg}$ cells were more sensitive to IL-4 polarization and concomitantly overexpressed Gata-3 and other $T_H2$-associated proteins compared to WT $T_{reg}$ cells. This was also observed upon in vivo infection with parasitic nematodes. Our

data showed that about two-times more $T_{reg}$ cells expressed high levels of Gata-3 in $\Delta/\Delta$ mice as compared to infected WT mice. As a consequence of Gata-3 expression, $\Delta/\Delta$ $T_{reg}$ cells differentiated into $T_H2$-polarized Klrg1+ST2+ tis$T_{reg}$ST2-like cells. This differentiation integrated a third critical restriction of $T_{reg}$ function and suppressive capacity: the loss of $T_H2$-suppressive capacity via the down-modulation of *Bach2*, *Dtx1*, and *Il2ra*, and the induction of ILT3. *Il2ra*, encoding for the IL-2 receptor alpha chain (CD25), is required for $T_{reg}$ suppressive capacity[40]. *Bach2* has been described as an important transcription factor required

**Fig. 5** Characterization of $T_{reg}$ isolated from WT vs. affected Δ/Δ animals. **a** Representative histograms of Foxp3, Ctla-4, and Helios expression in Δ/Δ $T_{reg}$ from affected animal (red dotted line), WT $T_{reg}$ (black line), and WT $T_{conv}$ (blue line). **b** Methylation of $T_{reg}$-specific demethylated region in $T_{reg}$ and $T_{conv}$ cells from WT and affected Δ/Δ animals. Yellow color indicates demethylated, blue methylated CpG dinucleotides. Total sequencing reads 2211 for WT $T_{conv}$, 67 for WT $T_{reg}$, and 129 for Δ/Δ $T_{reg}$. Statistical analysis one-way ANOVA and Bonferroni post-test ($n = 2$–6) in boxplots with center line (median), box (25–75 percentile) and standard deviation (min to max). **c** Pie charts illustrating TCR beta J-chain usage of $T_{reg}$ cells from WT vs. affected Δ/Δ animals. Colors indicate respective TCR beta J-chain. Individual mice are shown ($n = 3$). **d** Frequency of 10 most abundant $T_{reg}$ TCR sequences for each individual mouse from WT and affected Δ/Δ animals. All remaining sequences in blue. **e** Clonality (left) and entropy (right) values for all TCR sequences. Statistical testing unpaired $t$-test, individual mice are shown ($n = 3$). **f** Foxp3 staining in lung tissue sections from WT (left) or affected Δ/Δ animals (right). Additional Foxp3 stainings and controls in Supplementary Fig. 6. **g** Measurement of $T_{reg}$ cell in peripheral tissues. $T_{reg}$ (CD3$^+$CD4$^+$CD8$^-$TCRb$^+$ CD25$^+$Foxp3-GFP$^+$) percentage of CD4$^+$ (left) or total number (right) in respective tissue ($n = 5$, unpaired $t$-test). **h** Gata-3 protein staining of spleen-derived $T_{reg}$ cells (CD3$^+$CD4$^+$CD8$^-$CD25$^+$Foxp3$^+$). Pseudocolor plots representative examples, quantification of Gata-3-positive cells ($n = 9$–11) and *Gata3* mRNA expression of sorted $T_{reg}$ cells ($n = 4$–5) right (unpaired $t$-test). **i** Identification of T follicular regulatory cells in LN from WT vs. affected Δ/Δ animals. Tfr cells as CD3$^+$CD4$^+$CD8$^-$CD25$^+$Foxp3$^+$Cxcr-5$^+$PD-1$^+$ T cells and quantified right ($n = 4$, unpaired $t$-test). **j** CD3$^+$CD4$^+$CD8$^-$CD25$^+$Foxp3$^+$ $T_{reg}$ cells from LN of WT or affected Δ/Δ animals co-stained with PD-1 and Gata-3. Percentage of Gata-3$^+$PD-1$^+$ $T_{reg}$ cells quantified to the right ($n = 4$, unpaired $t$-test). Data are representative of two or more independent experiments with individual mice (**a**, **g**, **h**) or a single experiment with individual mice (**b**, **c**, **d**–**f**, **i**–**j**). Source data are provided as a Source Data file

to inhibit Gata-3 expression and $T_H2$ polarization[34,41], including the expression of ST2[42], and the Bach2–Batf interaction was shown to control $T_H2$-type immune responses[43]. In addition, Bach2 supports the suppressive capacity of $T_{reg}$ cells, as a loss-of-function study demonstrated that Bach2-deficient $T_{reg}$ cells failed to prevent disease in a colitis model[34]. *Dtx1*, previously reported to interact with Rbpj[32], was also shown to be important for $T_{reg}$ cell suppressive function in a transfer model of colitis[31]. Finally, the induction of ILT3 can lead to a severe defect in controlling $T_H2$-polarized immune responses via the induction of IRF4$^+$PD-L2$^+$ DCs[36].

This cumulative effect on $T_{reg}$ suppressive capacity was finally leading to a loss of $T_H2$ suppressive potential, a state where effector $T_H2$ cells produced more IL-4, leading to even more Gata-3 expression in Δ/Δ $T_{reg}$ cells. Gata-3 expression is required to maintain high Foxp3 expression levels and it is important to prevent $T_{reg}$ differentiation into an effector phenotype[44–46]. But Gata-3 does not function in a binary on–off mode. It has been reported that Gata-3 over-expression in T cell progenitors changes the identity of developing double-negative thymocytes and drives them into the mast cell lineage[47–49]. These studies indicate that a well-defined Gata-3 dosage is required for proper T cell development and function. A recent report showed that the IL-4 signaling strength is important for $T_{reg}$ cell function. By using mice carrying an IL-4Rα chain mutation leading to enhanced IL-4 signaling, the authors demonstrated that these $T_{reg}$ cells, which were polarized towards a $T_H2$ cell-like phenotype with high Gata-3 expression levels, had an impaired functionality[50].

Our data indicate that Rbpj could act as a Gata-3 dosage modifier, adjusting the balance of Gata-3 expression by restricting IL-4 sensitivity as a powerful molecular switch. In addition, published findings report that the *Gata3* gene itself is a direct target of Rbpj[18]. Therefore, the regulation of Gata-3 expression could be both on the transcriptional, as well as the IL-4 cytokine sensitivity level. Our findings should be considered in the current discussion that $T_H2$-polarized Gata-3$^+$ $T_{reg}$ cells are better suppressors of the corresponding $T_H2$-polarized effector T cells, a model proposed based on the complete deficiency of IRF4[13]. We could show that IL-4 and IL-33-induced Gata3$^{high}$ $T_{reg}$ cells express significantly more ILT3 upon in vitro expansion and differentiation, a surface receptor shown to interfere with efficient control of $T_H2$ effector cells[36]. Gata-3$^{high}$ $T_{reg}$ cells were unable to inhibit $T_H2$ differentiation of IL-4 exposed naive CD4 cells in vitro. In contrast to this, they supported the maturation of a $T_H2$-promoting IRF4$^+$PD-L2$^+$ DC subpopulation in vitro.

Our motif analysis of the ATAC-sequencing data revealed a strong Gata signature in Δ/Δ $T_{reg}$-specific differentially accessible regions. Many of the affected genes were shared with tis$T_{reg}$ST2 cells, a $T_H2$-biased $T_{reg}$ subset normally present within non-lymphoid tissues[7]. Rbpj may regulate the Gata-3-dependent $T_{reg}$ST2 differentiation pathway and, thereby, might limit the access to the $T_{reg}$ST2 compartment in lymphoid organs to allow the maintenance of a diverse $T_{reg}$ subset pool.

## Methods

**Mice**. Wildtype C57BL/6, congenic B6.SJL-Ptprc$^a$Pepc$^b$/BoyCrl (CD45.1$^+$), and congenic B6.PL-Thy1$^a$/CyJ (CD90.1$^+$) mice were obtained from Charles River Breeding Laboratories (Wilmington, MA, USA) or the Jackson Laboratory (Bar Harbor, ME, USA). B6N.129(Cg)-*Foxp3*$^{tm3Ayr}$ mice (*Foxp3*.IRES-DTR/GFP)[51] were bred to CD45.1$^+$ or CD90.1$^+$ mice in the animal facility of the German Cancer Research Center (DKFZ). B6.129(Cg)-*Foxp3*$^{tm4(YFP/cre)Ayr}$/J, Jackson (FOXP3.IRES-YFP/Cre)[52] were crossed to Rbpj$^{fl/fl}$ mice[53] to specifically delete Rbpj in $T_{reg}$ cells. Age-matched littermate controls (Foxp3$^{Cre,YFP}$-positive and wildtype for the Rbpj alleles) were used throughout the study. Details about hygiene status and barrier breeding conditions are explained in the following paragraph. Rag2-deficient (B6-Rag2tm1Fwa) lines were used to isolate protein for autoantibody detection. All animals were housed under specific pathogen-free conditions at the respective animal care facilities, and the governmental committee for animal experimentation (Regierungspräsidium Karlsruhe, Regierung von Unterfranken and Behörde für Gesundheit und Verbraucherschutz Hamburg) approved all experiments involving animals. Relevant ethical regulations for animal testing and research were complied with.

**Breeding conditions and disease-free survival in barriers**. Data from Fig. 1a are derived from animals housed under specified pathogen-free conditions in a specific mouse facility (called barrier 3) of the German Cancer Research Center, fulfilling the criteria given in the FELASA recommendations (animal number, health monitoring, age, agents, methods). All animals were housed in open cages allowing transmission of agents. Research personnel had access to the unit. Routine testing included testing for ectoparasites, endoparasites, bacteria and viruses. In barrier 3, murine norovirus (MNV), *Pneumocystis* sp. and *Staphylococcus aureus* have been detected, along with occasional detection of additional opportunistic agents. In our breeding, we observed 79 animals with $T_{reg}$ lineage-specific bi-allelic *Rbpj* deletion (*Foxp3*$^{Cre}$*Rbpj*$^{Δ/Δ}$), of which 21 were sacrificed due to sickness and used for experimentation. Twenty-nine animals <20 weeks old were otherwise healthy but used for experimentation and censored for survival analysis. Thirty-two animals grew older than 20 weeks and were marked healthy during the observation period, although some turned sick later and were used for experimentation. Out of the 67 animals with mono-allelic Rbpj deletion (*Foxp3*$^{Cre}$*Rbpj*$^{wt/Δ}$) and 79 wildtype animals (*Foxp3*$^{Cre}$*Rbpj*$^{wt/wt}$), 0 animals showed signs of disease. All *Foxp3*$^{Cre}$*Rbpj*$^{wt/Δ}$ and *Foxp3*$^{Cre}$*Rbpj*$^{wt/wt}$ were littermates of the *Foxp3*$^{Cre}$*Rbpj*$^{Δ/Δ}$ mice, housed in the same cages as their siblings.

To identify the influence of breeding conditions (and therefore the environmental influence) on disease development of *Foxp3*$^{Cre}$*Rbpj*$^{Δ/Δ}$ animals, we transferred the colony to a newly established mouse facility (called barrier A) via embryo transfer (results in Fig. 4a). Mice in this barrier were colonized with a defined Altered Schaedler Flora and are housed in individually ventilated cages. Access is limited to animal caretaker personnel. Until study end, the barrier was completely free of infectious agents listed in the FELASA recommendations. In this barrier, we observed 55 animals with $T_{reg}$ lineage-specific bi-allelic *Rbpj* deletion

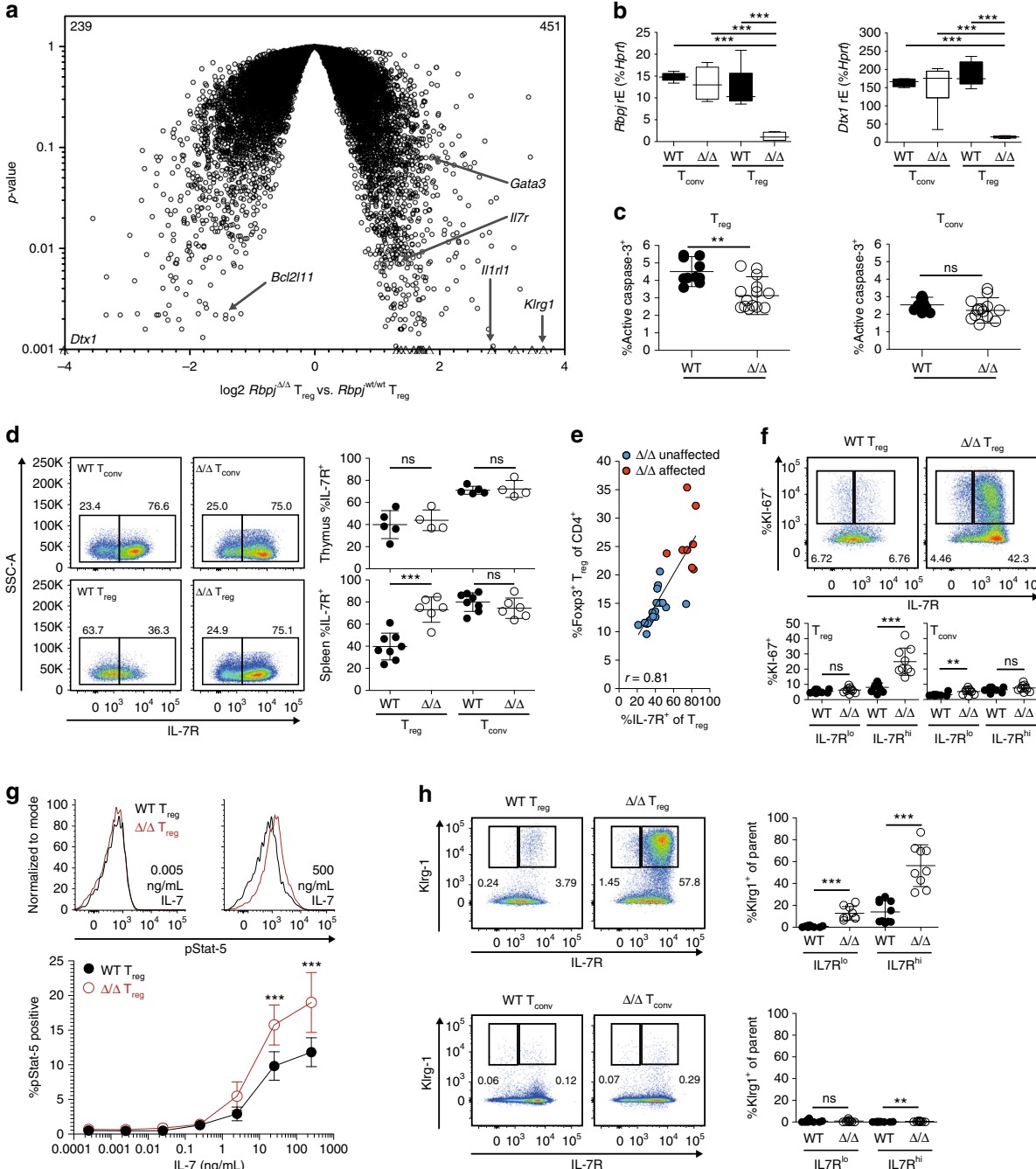

**Fig. 6** Profiling of Rbpj-deficient $T_{reg}$ cells. **a** Gene expression profile of $T_{reg}$ cells (CD3+CD4+CD8−CD25+Foxp3+) from spleens of WT vs. affected Δ/Δ animals. Selected genes labeled. Average values of three biological replicates shown. Numbers show significantly up-regulated or under-represented genes in the comparison ($p < 0.05$). Statistical calculations described in Methods. Genes with $p$-values of $< 0.001$ set to 0.001 and labeled with a triangle. **b** Measurement of *Rbpj* (left) and *Dtx1* (right) mRNA expression in FACS-isolated spleen-derived $T_{reg}$ and $T_{conv}$ cells from WT or unaffected Δ/Δ animals via Taqman qPCR ($n = 5$–6, one-way ANOVA with Newman–Keuls post-test). **c** Splenocytes isolated from WT and affected Δ/Δ animals treated with sulforhodamine-conjugated DEVD-FMK to label active caspase-3. $T_{reg}$ or $T_{conv}$ cells (CD3+CD4+CD8−CD25−Foxp3−) identified. Representative pseudocolor plots illustrating active Caspase-3 expression are shown in Supplementary Fig. 7. Statistical evaluation with unpaired $t$-test, $n = 10$–14. **d** Left panel, representative dot plots illustrating IL-7R expression on $T_{conv}$ cells (upper panel) and $T_{reg}$ cells (lower panel) found in WT (left) or affected Δ/Δ (right) animal-derived spleens; right panel, quantification ($n = 4$–8, unpaired $t$-test). **e** Correlation between frequency of IL-7R expression and frequency of $T_{reg}$ cells. Values for healthy unaffected Δ/Δ animals (blue dots) and affected Δ/Δ animals (red dots) shown ($n = 17$ for blue dots, and $n = 8$ for red dots). $R$ value displayed. **f** IL-7R and KI-67 expression in $T_{reg}$ cells or $T_{conv}$ cells derived from WT or affected Δ/Δ animals. Representative dot plots and quantification ($n = 9$–10, unpaired $t$-test). **g** Intracellular expression of phosphorylated Stat5 (pStat5) after IL-7 treatment of ex-vivo isolated splenocytes from WT vs. affected Δ/Δ animals. Representative histograms on top, percentage of intracellular pStat5 expression in $T_{reg}$ cells below. Statistics based on two-way ANOVA with Bonferroni post-test ($n = 6$). **h** IL-7R and Klrg-1 expression in $T_{reg}$ cells or $T_{conv}$ cells from WT or affected Δ/Δ animals. Representative pseudocolor plots and quantification for $T_{reg}$ and $T_{conv}$ cells ($n = 8$–10, unpaired $t$-test). Data are representative of two or more independent experiments with individual mice (**b–f**, **h**) or a single experiment with several individual mice (**a**, **g**). Source data are provided as a Source Data file

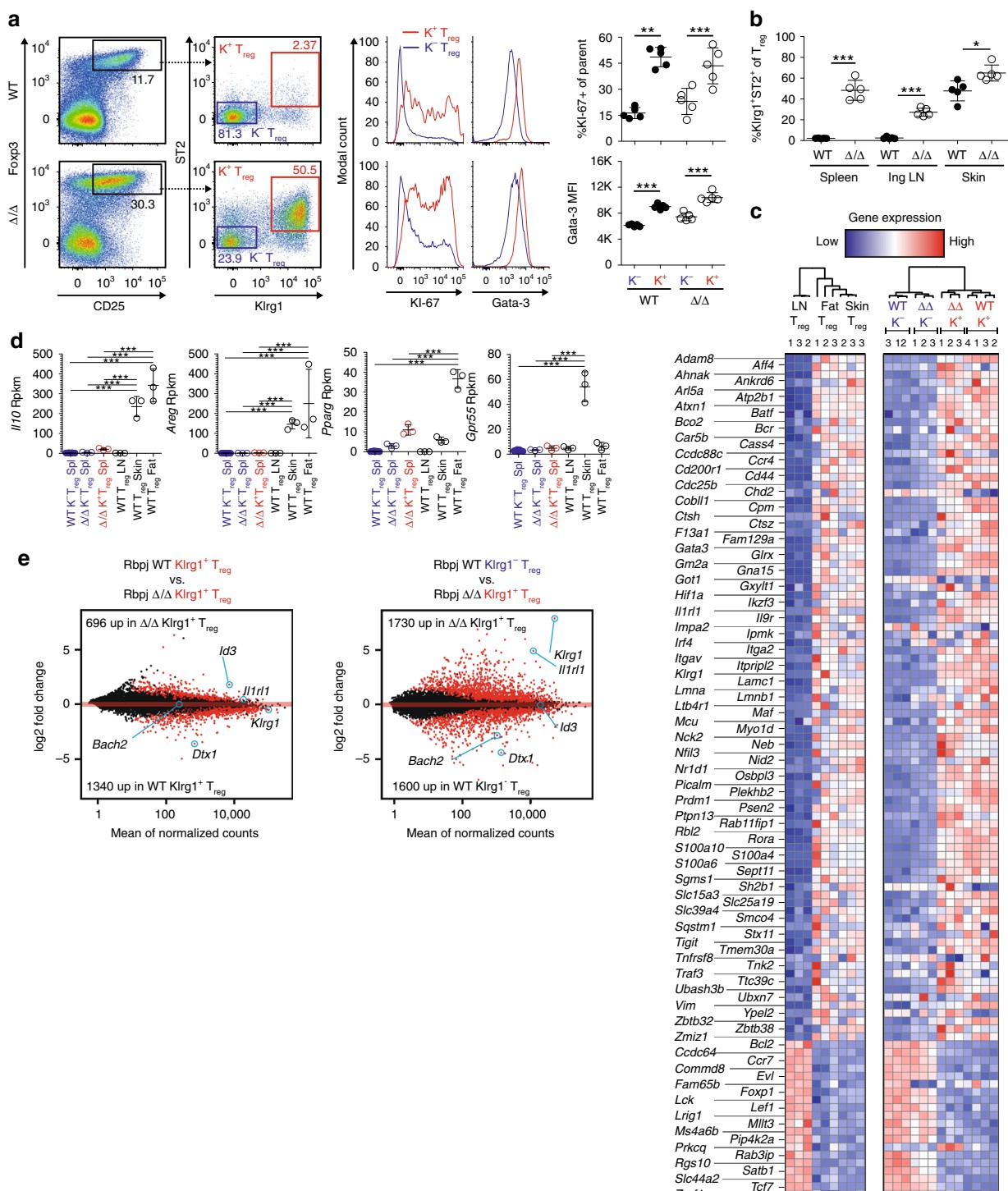

**Fig. 7** Enrichment of ST2+Klrg1+ Treg cells in affected Δ/Δ animals. **a** T_reg cells (CD4+CD45+CD25+Foxp3+) identified in spleens from WT vs. affected Δ/Δ animals and sub-gated for presence of ST2+Klrg1+ tisT_regST2-like cells (representative pseudocolor plots to the left). Middle, histograms illustrating KI-67 and Gata-3 expression in ST2+Klrg1+ (K+) vs. ST2−Klrg1− (K−) T_reg cells from WT and affected Δ/Δ animals, quantification to the right (n = 5, unpaired t-test). **b** Quantification of ST2+Klrg1+ T_reg cells in spleen, inguinal LN and skin of WT and affected Δ/Δ animals (% of T_reg, n = 5, unpaired t-test). **c** RNA sequencing from spleen Δ/Δ Klrg1+ T_reg cells (Δ/Δ K+ T_reg, red), Δ/Δ Klrg1− T_reg cells (Δ/Δ K− T_reg, blue), WT Klrg1+ T_reg cells (WT K+ T_reg, red), and WT Klrg1− T_reg cells (WT K− T_reg, blue). Expression of 106 tisT_regST2 signature genes derived from a previous study[7] in a heatmap with column dendrogram clustering. The expression of these signature genes in T_reg cells from fat, skin and LN T_reg cells shown to the left (Fat/Skin/LN WT T_reg). Heatmap and dendrogram created using R and heatmap function. **d** Expression of Il10, Areg, Pparg, and Gpr55 in WT T_reg cells from skin, fat, or LN versus spleen (spl) Δ/Δ Klrg1+ T_reg, spleen Δ/Δ Klrg1− T_reg, and spleen WT Klrg1− T_reg cells. Statistical analysis of RNA-seq data described in Methods section. **e** MA plots illustrating expression of genes in two comparisons. Left, spleen WT Klrg1+ T_reg cells vs. spleen Δ/Δ Klrg1+ T_reg cells; right, spleen WT Klrg1− T_reg cells vs. spleen Δ/Δ Klrg1+ T_reg cells. Individual genes are highlighted. Numbers indicate significantly expressed genes in the respective comparison. Data are representative of two or more independent experiments with individual mice (**a**, **b**) or a single experiment with several individual mice (**c–e**). Source data are provided as a Source Data file

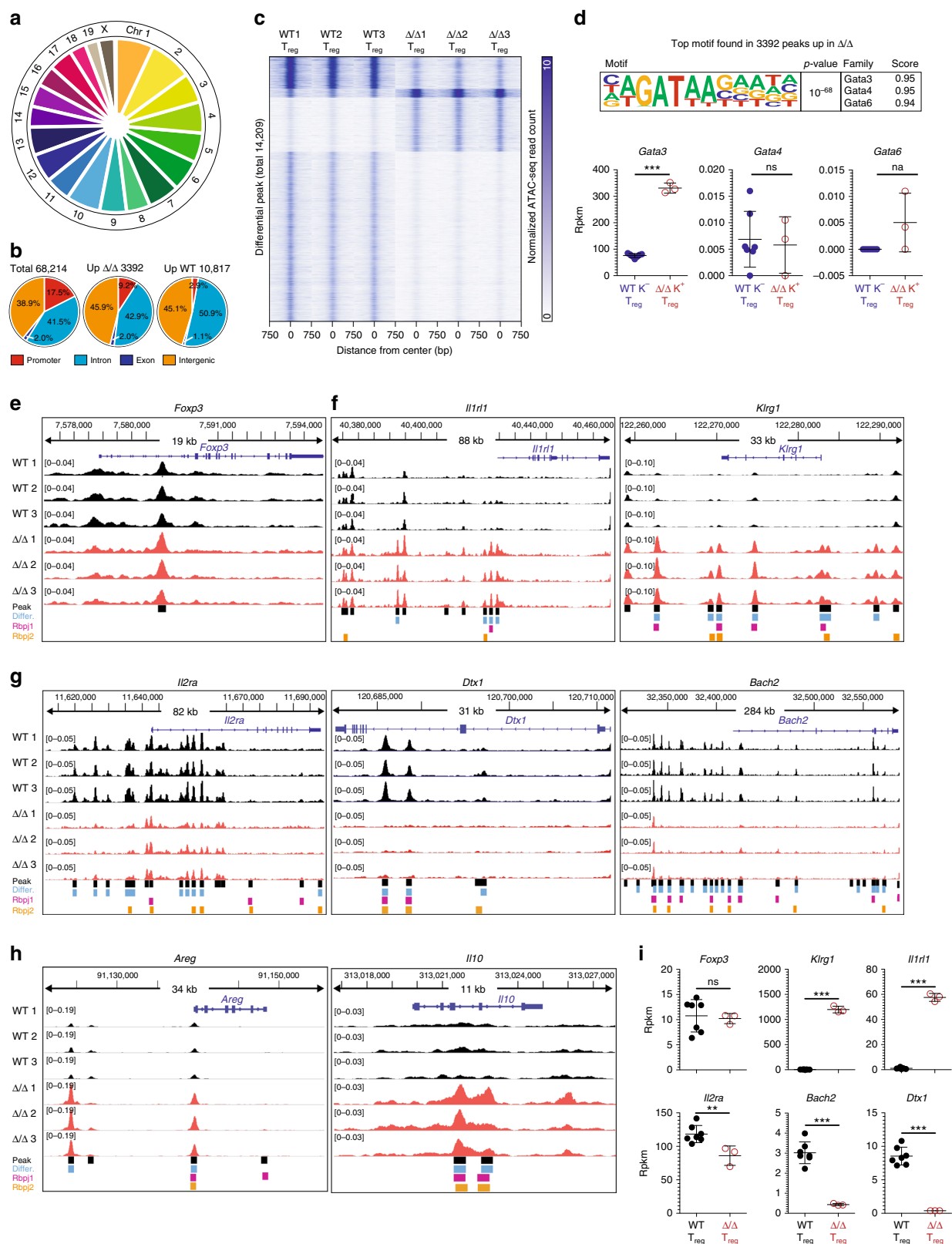

(Foxp3^Cre Rbpj^Δ/Δ), of which three were sacrificed due to sickness and used for experimentation. Seventeen animals <20 weeks old were otherwise healthy but used for experimentation and censored for survival analysis. Thirteen animals grew older than 20 weeks and were marked healthy during the observation period, although some turned sick later and were used for experimentation. Thirty-three animals were between 10 and 20 weeks old when the observation was stopped and the experiment concluded. Out of the 45 animals with mono-allelic Rbpj deletion

(Foxp3^Cre Rbpj^wt/Δ) and 40 wildtype animals (Foxp3^Cre Rbpj^wt/wt), 0 animals showed signs of disease.

**Rbpj genotyping.** Animal tails were digested in digest buffer (50 mM KCl, 20 mM Tris–HCl pH 8.8, 0.00045% Tween20 and Igepal CA-630) with proteinase K overnight at 56 °C. Following inactivation for 10 min at 96 °C, a PCR reaction with

**Fig. 8** ATAC-seq of Rbpj Δ/Δ $T_{reg}$ cells. **a** ATAC-seq with three biological replicates of Δ/Δ Klrg1$^+$ $T_{reg}$ cells and WT Klrg1$^-$ $T_{reg}$ cells. Identification of 68,214 peaks throughout the mouse genome. Distribution of all peaks across autosomes 1–19 and allosome x in a pie chart; pie size indicates contribution in percent. **b** Annotation of peaks to promoter (red), intron (light blue), exon (dark blue), or intergenic regions (orange). Left, annotation for all 68,214 peaks identified in the all dataset; middle, annotation for 3392 peaks specifically upregulated in Δ/Δ Klrg1$^+$ $T_{reg}$ cells; right, annotation for 10,816 peaks specifically upregulated in WT Klrg1$^-$ $T_{reg}$ cells. **c** Heatmap showing normalized ATAC-seq read counts in window of −750 bp to +750 bp around all differential peaks (14,208) for six samples. Y-axis individual peaks, X-axis distance from peak center. Color code indicates normalized ATAC-seq read count in 25 bp bins, with 0 = white and 10 = blue. **d** De novo motif analysis in 3392 peaks up in Δ/Δ $T_{reg}$ cells. Top enriched de novo motif is shown, along with the corresponding p-value and the three most-similar known motifs (similarity score from 0 to 1, with 1 indicating an exact match). Further motifs and motif analysis for the 10,816 peaks up in WT shown in Supplementary Fig. 8a, b. Below, gene expression of *Gata3*, *Gata4*, and *Gata6* (n = 3–7, unpaired t-test). **e–h** ATAC-seq genome browser tracks for eight genes, with WT Klrg1$^-$ $T_{reg}$ cell data in black and Δ/Δ Klrg1$^+$ $T_{reg}$ cell data in red. Gene information is shown on top, along with the genomic location. Height indicates normalized ATAC-seq signal, the scale shown in brackets. All samples are group-normalized to allow peak height comparison. Below, all peaks (black squares), differential peaks (blue squares), instances of de novo Rbpj-binding motif (purple) or literature-based Rbpj motif (orange) are shown. Displayed are: *Foxp3* (**e**), *Il1rl1* and *Klrg1* (**f**), *Il2ra*, *Dtx1*, and *Bach2* (**g**), *Areg* and *Il10* (**h**). **i** RNA expression values (Rpkm) for genes shown in (**e–g**), data derived from RNA sequencing (n = 3–7, unpaired t-test). Data are representative of experiments with several individual mice. Source data are provided as a Source Data file

Taq polymerase, dNTPs, 2.5 mM MgCl$_2$, and 0.4 µM *Rbpj* primers (GTGGAAC TTGCTATGTGCTTTG, CTGCCATATTGCTGAATGAAAA, CACATTCCCAT TATGATACTGAGTG) was started (95 °C, 5 min; 95°-30 s-58°-30 s-72°-30 s × 35; 72 °C, 5 min). Samples were separated on an agarose gel and gene status analyzed. Similarly, *Foxp3*-driven presence of *Cre* recombinase was detected via PCR (AGGA TGTGAGGGACTACCTCCTGTA, TCCTTCACTCTGATTCTGGCAATTT; 94 °C, 3 min; 94°-40s-60°-40s-72°-60 s × 30; 72 °C, 3 min).

**Flow cytometry and fluorescence-activated cell sorting**. Target organs were isolated and single-cell suspensions were established. If applicable, tissues were treated with collagenases and pre-purified according to protocol[7]. Red blood cells were lysed in hypotonic buffers, and cells were either pre-enriched with magnetic bead technology (Miltenyi Biotec) or directly stained with fluorochrome-labeled antibodies. Surface stainings were performed for 30 min at 4 °C, with all antibodies used at 1:100 dilution if not stated otherwise. If applicable, cells were fixed and afterwards permeabilized with the Foxp3 Fix/Perm Buffer set for 60 min at RT. Upon permeabilization, cells were stained intracellularly for 60 min at RT. Flow cytometry samples were acquired on a LSR II, Fortessa II or Canto II flow cytometer (BD Biosciences). For cell counting, AccuCheck counting beads were used (ThermoFisher). Samples for RNA isolation, DNA isolation, or subsequent cultivation were acquired and sorted on ARIA II or ARIA Fusion cell sorting systems (BD Biosciences) with four-way purity settings and an 85 µM nozzle. For RNA collection, cells were sorted in 500 µL of RNA lysis buffer, followed by RNA isolation based on manufacturer's instructions (RNEasy Mini Kit or RNEasy Micro Plus Kit, Quiagen). For DNA collection, samples were sorted into 500 µL of DNA lysis buffer and DNA was purified according to manufacturer's protocol (DNEasy Blood and Tissue Kit, Quiagen). To harvest protein, cells were sorted into FCS-containing buffer and afterwards pelleted. Cells were lysed in RIPA buffer.

**Real-time PCR**. For RNA isolation, antibody-labeled cells were sorted directly into RLT+lysis buffer and purified according to manufacturer's protocol (RNEasy mini Kit, Quiagen). RNA from whole tissues was isolated with mechanical tissue dissemination using ceramic beads followed by column-based RNA isolation (Innu-prep RNA Kit, Analytik Jena). RNA was reversely transcribed into cDNA according to manufacturer's protocol (Reverse Transcriptase II, life technologies). cDNA was used with Taqman probes and Taqman master mix or with Sybr primers and Sybr master mix in a Viia7 real-time PCR system (all ThermoFisher). Gene expression was normalized to housekeeping gene expression (*Hprt*) with the formula: relative gene expression = $2^{-(Ct\ (gene\ X)-Ct\ (Hprt))}$. Primer sequences and designations are listed in Supplementary Table 1.

**Detection of Rbpj protein via Western Blot**. $T_{reg}$ (CD3$^+$CD45$^+$CD4$^+$CD8$^-$ CD25$^+$Foxp3-GFP$^+$) and $T_{conv}$ cells (CD3$^+$CD45$^+$CD4$^+$CD8$^-$CD25$^-$Foxp3-GFP$^-$) were isolated via FACS. Cells were lysed in RIPA buffer and supplemented with Laemmli buffer containing beta-mercaptoethanol. Samples were then heated to 95 °C for 10 min and afterwards separated by SDS–PAGE with pre-cast gels (Biorad). Gels were blotted onto PVDF membranes according to standard protocol. Membranes were blocked with 5% Milk-PBST for one hour at RT followed by incubation over-night with an anti-mouse Rbpj primary monoclonal antibody (Cell signaling clone D10A4) at 1:3000 in 5% Milk-PBST. Membranes were washed and RBPJ antibody was labeled with an HRP-conjugated secondary antibody at 1:10,000 dilution for one hour at RT. Membranes were washed and developed with a chromogenic detection reagent (Thermo Fisher).

**Histology and microscopy**. Immediately after the animals were sacrificed, organs were carefully excised and stored overnight in freshly prepared 4% formaldehyde solution. Afterwards, samples were embedded, thin-cut (thickness between 3 and 5 µM) and stained. Haematoxylin and eosin (H&E) stainings, periodic acid-Schiff

reaction (PAS) stainings, and Giemsa stainings were prepared according to literature[54]. Representative images were acquired on a Zeiss Axioplan microscope equipped with a AxioCam ICc 3 color camera with ZEN 2011 lite (BLUE EDITION) software. Intensity and contrast settings were adjusted for each organ but kept consistent between control and test sample. Foxp3-staining on embedded tissues was performed as follows: first, samples were thin-cut (3–5 µm); second, paraffin was melted (72 °C, 30 min) followed by Xylol (2 × 5 min), and alcohol treatment; third, samples were incubated at 120 °C in Tris–EDTA buffer for 5 min, followed by blocking with peroxidase-block (Dako); then, incubation with primary antibody (Foxp3 FJK-16 s, 1:50, 30 min at RT) and secondary antibody (anti-rat HRP, 30 min at RT) with intermittent washing steps for 5 min (Washing buffer from Dako); last, chromogenic detection solution was added for 10 min at RT (DAB+ substrate, Dako) followed by 1 min incubation with Hematoxylin (Merck) and washing steps.

**Immunohistochemistry and slide scanning/image analysis**. Foxp3 stainings of embedded tissues were prepared with a fluorescence-labeled anti-Foxp3 antibody as described in literature[55]. To visualize germinal centre formation, lymph node samples were recovered from animals, immediately frozen in TissueTec buffer (Sakura Fineteck Europe) on cold carbon dioxide pellets and stored at −80 °C. Individual samples were cut from the tissue block and fixed with acetone for 10 min. Following blocking with 10% FCS, slides were incubated with AF488-labeled anti-mouse GL7 antibody (1:20), AF594-labeled anti-mouse IgD antibody (1:20), and AF 647-labeled anti-mouse CD4 antibody (1:20) overnight at 4 °C. After washing, samples were mounted with a fluorescent mounting medium (Dako). Unstained or single-stained samples were prepared and imaged as background staining controls. Samples were imaged on a motorized Zeiss inverted Cell Observer.Z1 with a mercury arc burner HXP 120/Colibri LED module, as well as a gray scale CCD camera AxioCam and a color CCD camera AxioCam MRc. Images were sequentially scanned and assembled with ZEN 2011 lite (BLUE EDITION) software. In order to obtain full-size images of lymph nodes, we first scanned the green channel (AF488, GL-7) followed by red channel (AF594, IgD), and blue channel (AF647, CD4). Contrast and fluorescence intensity were adjusted for all samples in parallel. To reduce robotic scanning errors, images were stitched to smooth transition areas. For Foxp3 stainings, staining intensity was normalized for each image.

**Gene expression analysis with bead chips and statistics**. For whole lymph-node gene expression analysis, inguinal lymph nodes from WT and affected Δ/Δ animals underwent mechanical tissue dissemination using ceramic beads followed by column-based RNA isolation, as described earlier. For gene expression analysis of T cells, we FACS-isolated spleen-derived $T_{reg}$ (CD3$^+$CD4$^+$CD8$^-$CD45$^+$CD25$^+$Foxp3-YFP$^+$) and $T_{conv}$ cells (CD3$^+$CD4$^+$CD8$^-$CD45$^+$CD25$^-$Foxp3-YFP$^-$) from WT and Δ/Δ animals. RNA was isolated (RNEasy mini kit) and the DKFZ Genomics and Proteomics Core Facility amplified and hybridized material to the Illumina MouseWG-6 v2.0 Expression BeadChip. Microarray scanning was done using an iScan array scanner. Data extraction was done for all beads individually, and outliers were removed when the absolute difference to the median was >2.5 times MAD (2.5 Hampelís method). Expression values were quantile normalized and log2-transformed. Differentially expressed probes between groups were identified using the empirical Bayes approach based on moderated t-statistics as implemented in the Bioconductor package limma. All p-values in limma were adjusted for multiple testing using Benjamini–Hochberg correction in order to control the FDR. Probes with an FDR < 5% were considered statistically significant.

**Statistical analysis of data**. Data were analyzed with Prism software. We used a log-rank Mantel–Cox test in Kaplan–Meier survival curves (Figs. 1a and 4a), Mann–Whitney testing (Figs. 1b, c, d; 2b, c; 3a, b), unpaired t testing (Figs. 3d, e, f;

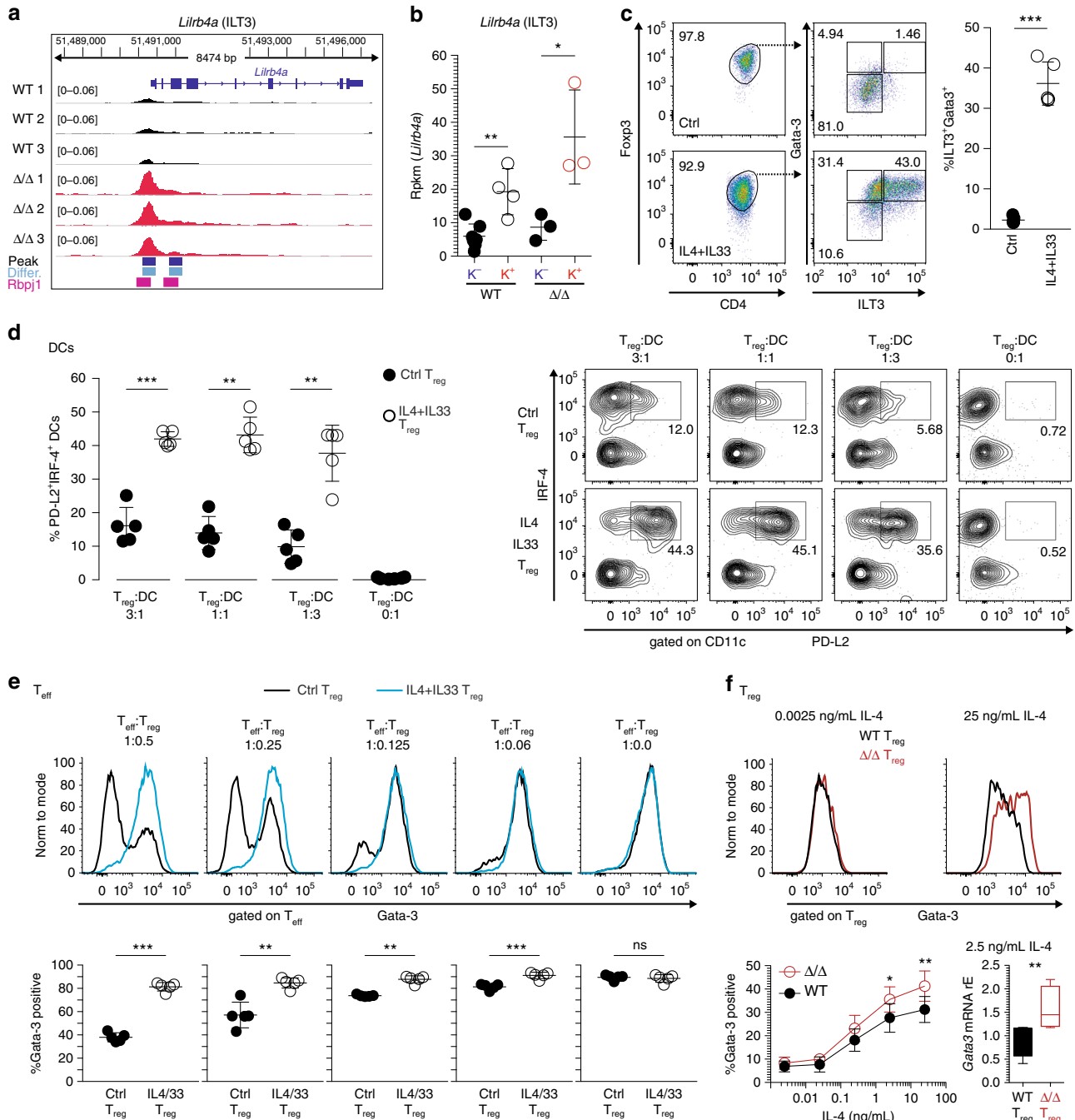

**Fig. 9** ILT3+Gata3+ T_H2-biased T_reg cells fail to suppress T_H2 polarization. **a** ATAC-seq genome browser track for the *Lilrb4a* gene (encoding for ILT3), with WT Klrg1−T_reg cell data in black and Δ/Δ Klrg1+ T_reg cell data in red. Gene information on top. Height indicates normalized ATAC-seq signal, scale shown in brackets. All samples group-normalized. Below, all peaks (black squares), differential peaks (blue squares), and de novo Rbpj-binding motif (purple). **b** *Lilrb4a* gene expression based on RNA-seq data derived from WT and Δ/Δ Klrg1− and Klrg1+ T_reg populations (n = 3–7, unpaired t-test). **c** WT T_reg cells (CD4+CD25+Foxp3-GFP+) expanded for 6 days in vitro with IL-4 and IL-33 (IL4+IL33 T_reg group) or without (control, Ctrl T_reg group). After 6 days, expression of Foxp3, Gata-3, and ILT3 measured by flow cytometry. Quantification right (n = 4, unpaired t-test). **d** 16-h DC polarization assay with different numbers of Gata-3-expressing T_reg cells or control T_reg cells (Ctrl T_reg) as in **c**. Expression of PD-L2 and IRF-4 in DCs (CD11c+MHCII+) measured by flow cytometry. Quantification left (paired t-test, n = 5) and representative dot plots right. **e** 96-h T-effector cell (T_eff) polarization assay with 20 ng/mL IL-4 and different numbers of either Gata-3-expressing T_reg cells (blue line) or control T_reg cells (black line) as in **c** and **d**, in the presence of DCs. Representative histograms showing Gata-3 expression in T_eff cells (CD4+CD11c-MHCII−CD25+Foxp3−) on top, quantification below (n = 5, paired t-test). **f** T_reg cells (CD4+CD25+Foxp3-YFP+) isolated from healthy, unaffected Δ/Δ and WT animals and stimulated with escalating doses of IL-4 followed by FACS measurement of Gata-3 protein. Representative histograms on top show Gata-3 expression in WT (black) or Δ/Δ (red) T_reg cells, treated with 0.0025 ng/mL IL-4 (left) or 25 ng/mL IL-4 (right). Below, Gata-3 expression across different doses. Statistics based on two-way ANOVA with Bonferroni post-test (n = 6). Lower right, qPCR-based verification of flow cytometry data for *Gata3* with 2.5 ng/mL IL-4 (n = 4, paired t-test). Data are representative of two or more independent experiments with individual mice (**c**–**f**) or a single experiment with several individual mice (**a**, **b**). Source data are provided as a Source Data file

4c, d, f, g, i, j; 5e, g–j; 6c, d, f, h; 7a, b; 8d, i; 9b, c, Supplementary Fig. 1a, c, d; Supplementary Fig. 2a), paired t testing (Figs. 4h and 9d–f), one-way ANOVA with Bonferroni post-testing (Figs. 5b and 6b; Supplementary Fig. 1b, Supplementary Fig. 2b) or Newmann–Keuls post-testing (Fig. 6b), or two-way ANOVA with Bonferroni post-testing (Figs. 6g and 9f, Supplementary Fig. 3). RNA-sequencing data in Figs. 2a, 6a and 7d, e were statistically evaluated as described in the respective paragraph of the Methods section. The respective number of animals (n) as well as the statistical test is also listed in the figure legend. All statistical test results and data used to calculate statistics are incorporated into the source data file.

**Blood serum Ig subtype analysis**. Blood was extracted via cardiac puncture from sacrificed animals and allowed to clot for at least 15 min. Afterwards, samples were centrifuged at $13,000 \times g$ for 15 min and serum was collected. ELISA plates (Costar #9018) were pre-coated with goat-anti-mouse IgG + IgM (Dianova #115-005-0068) or goat-anti-mouse IgE (Biozol #1110-01) overnight at 4 °C, followed by washing and blocking (PBS 0.2% gelatine 0.1% NaN₃). Wells were incubated with serial dilutions of serum or control antibody for one hour at RT, followed by four washing steps. Peroxidase-conjugated secondary antibodies were added at 1:1000 dilution in PBS and incubated for one hour at RT, again followed by four washing steps. Plates were developed with 1 mg/mL OPD in 0.1 M $KH_2PO_4$ (pH 6.0) solution with 1 μL/mL of 30% $H_2O_2$ solution. Once colorimetric reaction was complete, incubation was stopped with 25 μL 1 M $H_2SO_4$ and read on a ELISA photometer at 490 nm wave length. All antibodies are listed in Supplementary Table 1.

**Isolation of blood plasma and blood serum**. Blood was collected from sacrificed mice via cardiac puncture. Blood was mixed with Heparin–PBS to a final concentration of 20 U/mL Heparin. Samples were centrifuged at $3000 \times g$ for 15 min at 4 °C. Blood parameters were measured by photometric analysis on the ADVIA 2400 system (Siemens Healthcare Diagnostics) in the Zentrallabor (Medical Clinic-1, Analysezentrum) of the Heidelberg University Clinic. For blood serum collection, blood was extracted via cardiac puncture and allowed to clot for at least 15 min. Afterwards, samples were centrifuged at $13,000 \times g$ for 15 min and blood serum was collected.

**Autoantibody detection via Western Blot**. Following organs and tissues from RAG2⁻/⁻ animals were isolated: brain, eye, spleen, lymph nodes, pancreas, salivary gland, stomach, liver, kidneys, heart, lung, testis, small, and large intestine. Tissues were weighted and adjusted to 100 mg, followed by addition of 500 μL ProteoJET mammalian cell lysis reagent (Fermentas) supplemented with 2X protease inhibitor (Roche Diagnostics). Tissues were mechanically dissected using scissors and incubated for 10 min at RT on an orbital shaker. Samples were then centrifuged for 15 min at $16,000 \times g$ and supernatant containing protein lysate was harvested. Protein concentration was determined by BCA assay (Pierce). Twenty micrograms of protein were loaded per well on a pre-cast SDS gel (Biorad), followed by gel electrophoretic separation. Each gel contained eight wells with the same protein load (e.g. pancreas). After transfer on a PVDF membrane, this membrane was cut into eight strips containing the protein of interest. PVDF membranes were blocked with 5% Milk–PBST solution for one hour at RT following incubation with peripheral blood serum of individual mice (4 WT, 4 Δ/Δ) overnight at 4 °C (concentration 1:500 in 5% Milk–PBST). Individual strips were washed three times with PBST, followed by incubation with a HRP-conjugated donkey-anti-mouse IgG antibody (concentration 1:3000). After washing, strips were re-assembled and HRP activity was detected with a chromogenic substrate (Thermo Fisher).

**Infection of WT and Δ/Δ animals with *S. ratti***. Animal experimentation was conducted at the animal facility of the Bernhard Nocht Institute for Tropical Medicine in agreement with the German animal protection law under the supervision of a veterinarian. The experimental protocols have been reviewed and approved by the responsible federal health authorities of the state of Hamburg, Germany, the Behörde für Gesundheit und Verbraucherschutz. Mice were sacrificed by cervical dislocation under deep CO₂ narcosis. Two cohorts with five WT and five healthy Δ/Δ animals each were infected on day 0 by injection of 1000 *S. ratti* larvae subcutaneously into the footpad. Cohort 1 was sacrificed 6 days after infection and intestinal parasite count, stool PCR, and flow cytometry (Treg frequency, cytokine restimulation) were performed. Cohort 2 was sacrificed 14 days after infection; stool collection on day 6, 8, 10, 14 followed by stool-PCR; flow cytometry (Treg frequency, cytokine restimulation) on day 14. To count the number of adult parasitic females in the gut, the small intestine was flushed slowly with tap water to remove feces, sliced open longitudinally and incubated at 37 °C for 3 h in a Petri dish with tap water. The released adult females were collected by centrifugation for 5 min at 1200 rpm and counted. To quantify the release of *S. ratti* larvae by infected mice, the feces of individual mice was collected over 24 h periods and DNA from representative 200 mg samples was extracted as described[56]. Two hundred nanograms of DNA was used as a template for qPCR specific for *S. ratti* 28 S ribosomal RNA gene as described[57]. For analysis of serum antibodies, blood was collected from infected mice at the indicated time points and allowed to coagulate for 1 h at RT. Serum was collected after centrifugation at $10,000 \times g$ for 10

min at RT and stored at −20 °C for further analysis. *Strongyloides*-specific IgM in the serum was quantified by ELISA, as described[57]. Serum concentration of IgE was quantified using the IgE ELISA kit according to the manufacturers recommendations. In flow cytometry experiments, spleens were collected, red blood cells lysed and samples stained as described previously. For cytokine restimulation experiments, single-cell suspensions were incubated with cell stimulation cocktail plus transport inhibitor for 6 h at 37 °C followed by intracellular cytokine staining. To measure secreted IL-9 levels, mesenteric LN-derived cells were mashed and incubated with anti-CD3 antibody (1 μg/mL) for 72 h at 37 °C with $1 \times 10^6$ cells per well. IL-9 levels were quantified using the IL-9 ELISA kit according to the manufacturers recommendations.

**Methylation of the TSDR**. Genomic DNA of sorted cell populations was purified according to manufacturer's guidelines using the DNEasy Blood and Tissue kit (Quiagen). DNA purity and concentration were measured with a NanoDrop® photometer. Bisulfite-conversion was performed using the EpiTect Bisulfite Conversion Kit (Quiagen). Barcode-labeled primers for the *Foxp3* CNS2 (TSDR) were used to generate PCR amplicons from bisulfite-converted DNA (ForP: TGGGTT TTTTTGGTATTTAAGAAAG; RevP: AAAAAACAAATAATCTACCCCACAA). PCR amplicons were separated from primer dimers on a 2% agarose gel and purified using a Quick Gel Extraction Kit (Life Technologies). PCR amplicons were processed on a GS Junior Sequencer (Roche). Sequence reads were aligned to the BS-converted mouse genome and visualized.

**In vitro Treg suppression assay**. First, we isolated MHCII-positive antigen-presenting cells (CD8⁻CD25⁻Foxp3-GFP⁻MHCII⁺CD90.1⁻) as well as CD4-positive T-responder cells (CD4⁺CD8⁻CD25⁻Foxp3-GFP⁻MHCII⁻CD90.1⁺ CD90.2⁻) from *Foxp3*GFP, CD90.1 mice. T-responder cells were then labeled with carboxyfluorescein diacetate succinimidyl ester (CFSE) at 1 μM concentration in 10 mL cell culture medium for 15 min at RT, followed by several washing steps. Next, Treg (CD4⁺CD8⁻CD25⁺Foxp3-YFP⁺MHCII⁻CD90.1⁻CD90.2⁺) and Tconv (CD4⁺ CD8⁻CD25⁻Foxp3-YFP⁻MHCII⁻CD90.1⁻CD90.2⁺) cells from WT and affected Δ/Δ mice were isolated by FACS and serially diluted. In each well, 50,000 T-responder cells, 100,000 MHCII-positive APCs, and serially diluted Treg or Tconv cells were added. To stimulate APC-driven T-responder cell proliferation, 2 μg/mL anti-CD3 antibody was added (Biolegend clone OKT3). Cells were incubated for 5 days at 37 °C, followed by re-staining for flow cytometric analysis of CFSE-dye dilution in T-responder cells.

**TCR sequencing**. Single cell suspensions from spleen and lymph nodes (combined axial, cervical, brachial) from individual WT vs. Δ/Δ mice were established, and red blood cells lysed. Treg cells (CD3⁺CD4⁺CD8⁻CD45⁺CD25⁺Foxp3-YFP⁺) were pre-enriched with CD25-magnetic bead-based purification and sorted via FACS. Genomic DNA was isolated with the DNEasy blood and tissue kit according to manufacturer's instructions and measured on nanodrop photometer. Five hundred nanograms of gDNA per individual mouse were shipped to Adaptive Biotechnologies (Seattle, WA) for TCR sequencing. Data were analyzed with online tools provided by Adaptive Biotechnologies.

**Active caspase-3 assay and intracellular cytokine secretion**. For measurement of active caspase-3, single-cell suspensions were resuspended in FCS-containing cell culture medium (Gibco) and 1 μL of Red-DEVD-FMK (abcam) was added to each well in a 96-well tissue culture plate. Samples were incubated for 60 min at 37 °C, followed by washing with supplied wash buffer and surface antibody staining. As a positive control, splenocytes were incubated for 5 min at 42 °C. To measure intracellular cytokines, splenocytes, or lymph node-derived cell suspension were resuspended in FCS-containing cell culture medium. For stimulation, cells received 1X PMA-Ionomycin cocktail plus transport inhibitor, whereas controls received 1X transport inhibitor only cocktails (eBiosciences). Samples were incubated for 6–8 h at 37 °C, followed by surface and intracellular staining with Foxp3 Fix/perm staining mix.

**Detection of phosphorylated Stat5 or Stat6**. Spleen-derived single-cell suspensions were resuspended in FCS-containing cell culture medium (Gibco) and incubated at 37 °C in a 96-well tissue culture plate. Then, mouse recombinant IL-2, IL-4, or IL-7 were added in a 1+9 titration curve with a pre-warmed 2X mixture to cell suspension. Cells were incubated with the respective cytokine for 10 min at 37 °C. Afterwards, cells were centrifuged at $1000 \times g$ and 4 °C for 2 min, followed by two washing steps with cold FACS buffer. Cells were fixed with 1X Fixation buffer (BD Fixation Buffer) for 30 min at 4 °C. Afterwards, cells were centrifuged and resuspended in −20 °C cooled Perm Buffer (BD PermBuffer III) and incubated for 30 min at 4 °C. Afterwards, cells were washed twice and stained with surface and intracellular antibodies for 30 min at 4 °C, followed by analysis via flow cytometry.

**RNA sequencing**. cDNA was generated and amplified using 4.8 ng of total RNA (RNEeasy Mini Kit) and SMARTer Ultra Low Input RNA for Illumina Sequencing —HV (Clontech Laboratories, Inc.) according to the manufacturer's protocol. Then, sequencing libraries were prepared using the NEXT ChIP-Seq Library Prep Master Mix Set for Illumina (New England Biolabs) according to the

manufacturer's instructions with the following modifications: The adapter-ligated double-stranded cDNA (10 μL) was amplified using NEBNext Multiplex Oligos for Illumina (New England Biolabs, 25 μM primers), NEBNext High-Fidelity 2x PCR Master Mix (New England Biolabs) and 15 cycles of PCR. Final libraries were validated using Agilent 2100 Bioanalyzer (Agilent Technologies) and Qubit flourometer (Invitrogen), normalized and pooled in equimolar ratios. 50 bp single-read sequencing was performed on the Illumina HiSeq 2000 v4 according to the manufacturer's protocol.

**Mapping of RNA seq data, statistical evaluation, and plotting**. For all samples, low-quality bases were removed with Fastq_quality_filter from the FASTX Toolkit 0.0.13 (http://hannonlab.cshl.edu/fastx_toolkit/index.html) with 90% of the read needing a quality phred score >20. Homertools 4.7[58] were used for PolyA-tail trimming, and reads with a length <17 were removed. PicardTools 1.78 (https://broadinstitute.github.io/picard/) were used to compute the quality metrics with CollectRNASeqMetrics. With STAR 2.3[59], the filtered reads were mapped against mouse genome 38 using default parameters. Count data and RPKM tables were generated by mapping filtered reads against union transcripts (derived from Mouse Ensembl 90) using a custom pipeline. Mapping was carried out with bowtie2 version 2.2.4[60] against union mouse genes: every gene is represented by a union of all its transcripts (exons). The count values (RPKM and raw counts) were calculated by running CoverageBed from Bedtools v2.26.0[61] of the mapped reads together with the mouse annotation file (Ensembl 90) in gtf format and parsing the output with custom perl scripts. The input tables containing the replicates for groups to compare were created by a custom perl script. For DESeq2[62], DESeq-DataSetFromMatrix was applied, followed by estimateSizeFactors, estimateDispersions, and nbinomWald testing. The result tables were annotated with gene information (gene symbol, gene type) derived from the gencode.vM16.annotation.gtf file. The results were then filtered for protein-coding genes according to the gencode.vM16.annotation.gtf file. To account for the gender differences between male and female mice, 4571 differently expressed genes with an adj. $p$-value < 0.05 in the comparison of wildtype spleen-derived TCRβ$^+$CD4$^+$CD8$^-$CD25$^+$Foxp3 (GFP)$^+$Klrg1$^-$ST2$^-$ (female) with wildtype spleen-derived TCRβ$^+$CD4$^+$CD8$^-$CD25$^+$Foxp3(GFP)$^+$Klrg1$^-$ST2$^-$ (male) cells were excluded from the input tables into DESeq2 and the analyses were rerun with the remaining genes. MA plots were generated as described in ref. [63]. PCA plots and tables were generated using the plotPCA function of DESeq2 for the most variable 500 genes after applying the DESeq2 varianceStabilizingTransformation to the data.

**ATAC-seq**. The assay for transposase-accessible chromatin using sequencing (ATAC-seq) was performed according to Buenrostro et al. [35] with some modifications. In brief, about 50,000 FACS-isolated cells were pelleted on with 10,000 × g for 3 min and supernatant removed. Cells were tagmented at 55 °C for 8 min in 50 μL 1x tagmentation buffer including 2.5 μL transposome from the Nextera DNA library prep kit and 0.01% digitonin. The transposome was inactivated by addition of 20 μL 5 M guanidinium thiocyanate, and the DNA was purified with two volumes, i.e., 140 μL, of DNA-binding beads (HighPrep beads). The DNA was PCR amplified with a LightCycler 480 (Roche) in a 50 μL reaction with the NEBNext High Fidelity mix including 0.5 μL 1x SYBRGreen, forward primer Tn5McP1n (AATGATACGGCGACCACCGAGATCTACACTCGTCGGCAGCGTC) and barcoded reverse primers Tn5mCBar (CAAGCAGAAGACGGCATACGAGAT (8–9N barcode) GTCTCGTGGGCTCGG); 72 °C, 5 min; 98 C, 30 s (gap repair and initial melting); then cycling with 98 °C, 10 s; 63 °C, 30 s; 72 °C, 30 s for 15 cycles when all amplifications turned into saturation. PCR products were purified with 1.4 volumes (70 μL) magnetic beads. Ten microliters of eluates were sequenced on a HiSeq2000, 125 bp paired-end.

**ATAC-seq data analysis**. ATAC-seq read data were processed as described previously[64]. Reads were trimmed using Skewer[65] and aligned to the mm10 assembly of the murine genome using Bowtie2[60] with the '-very-sensitive' parameter. Duplicate reads were removed using sambamba markdup[66], and only properly paired reads with mapping quality >30 were kept. Reads were shifted as previously described to account for the transposition event[35]. Bigwig files were created using bedtools[61]. Preprocessed ATAC-seq data was analyzed with the HOMER package[58]. In brief, mapped sequencing reads were transformed to tag directories using the option '-sensitivity 3'. Peaks were called on all reads in all samples using the 'findPeaks' command with the options '-style factor -size 350 -minDist 300 -L 2'. Peaks from the mitochondrial genome and ENCODE blacklisted regions[67] were removed. Based on this consensus peak set, differential peaks were called using the DEseq2 implementation of HOMER. Specifically, reads form all individual experiments were counted in all individual peaks of consensus peak set using the 'annotatePeaks.pl' command with the option '-raw'. Based on the read count at each peak, significantly differential peaks were called using the 'getDiffExpression.pl' command using the '-norm2total' option. Motifs were then called on differential peaks using the 'findMotifsGenome.pl' command using the 'hypergeometric' option for $p$-value calculations. As a background, the non-differential peak set was used. The final motif sets were filtered for motifs with a significance <1e−10 and with an occurrence of <0.5% in the background regions set in case the non-differential peaks were used as a background file for motif finding using the

'compareMotifs.pl' command. To create heatmaps of differential peaks, sequencing reads were annotated to the differential peaks in 25 bp bins in a region of −750 bp to +750 bp from the peak center. The data matrix was clustered using Cluster 3.0[68] with the $k$-means option, and the heatmap was visualized with the TreeView 3[69] software.

**In-vitro T$_{reg}$ differentiation with IL-4 and IL-33**. Spleen and lymph-nodes were isolated and single-cell suspensions established. T$_{reg}$ cells were pre-purified using CD25-based magnetic bead staining and column-based isolation, followed by FACS-based sorting of CD4$^+$CD25$^+$Foxp3-GFP$^+$ T$_{reg}$ cells. 20,000 cells each were supplemented with either recombinant mouse IL-4 (500 ng/mL) plus recombinant mouse IL-33 (500 ng/mL) (IL4 + IL33 T$_{reg}$ group) or without additional cytokines (Crtl T$_{reg}$ group). In addition, both T$_{reg}$ groups were cultured with 5000 U/mL IL-2 and anti-CD3/28 microbeads at 4:1 bead to cell ratio were added to each well. Cells were incubated with the respective cytokine mix for 6 days at 37 °C, medium was re-supplemented on day 4. On day 6, cells were counted using a flow cytometer and expression of ILT3 and Gata-3 was measured. Cells were washed three times to remove remaining cytokines and used for the in vitro DC differentiation assay and for the in vitro T$_{eff}$ polarization assay.

**In vitro DC differentiation assay**. Spleen and mesenteric LNs from donor animals were isolated and single-cell suspensions established. DCs were pre-purified using CD11c microbeads and column-based isolation, followed by FACS-based sorting of a MCHII$^+$CD11c$^+$ population. DCs were stimulated with LPS (100 ng/mL), and anti-CD3 (4 μg/mL) was added. In vitro expanded T$_{reg}$ cells were added at indicated ratios (APC:T$_{reg}$ 1:3, 1:1, 3:1, 1:0) and incubated for 16 h at 37 °C. Cells were fixed, and DCs were analyzed after antibody surface and intracellular staining with the BD Fix/Perm Buffer kit.

**In vitro T$_{eff}$ polarization assay**. Spleen and mesenteric LNs from donor animals were isolated and single-cell suspensions established. T$_{eff}$ cells were pre-purified using CD4 microbeads and column-based isolation, followed by FACS-based sorting of CD4$^+$CD25$^-$Foxp3-GFP$^-$CD62L$^+$ naïve T$_{eff}$ cells. 5 × 10$^4$-purified T$_{eff}$ cells were co-cultured with expanded T$_{reg}$ cells at various ratios (T$_{eff}$:T$_{reg}$ 1:0.5, 1:0.25, 1:0.125, 1:0.06, 1:0) and 1 × 10$^4$ FACS-sorted DCs (MCHII$^+$CD11c$^+$). Anti-CD3 (4 μg/mL) and recombinant mouse IL-4 (20 ng/mL), as well as mouse IFNγ-blocking mAb (20 μg/mL) were added and cells were incubated for 96 h at 37 °C. Cells were fixed, followed by surface and intracellular staining with the Foxp3 Fix/Perm Buffer kit.

**Measurement of Gata-3 induction upon cytokine challenge**. Spleen and lymph-node-derived T$_{reg}$ cells (CD4$^+$CD8$^-$CD25$^+$Foxp3-YFP$^+$) were isolated from WT and healthy Δ/Δ animals via FACS. 75,000 cells each were supplemented with IL-4 at different concentrations, along with 5000 U/mL IL-2, 20 μg/mL IL-12 blocking mAb, 20 μg/mL IFNγ-blocking antibody, and CD3/CD28 microbeads at 4:1 bead to cell ratio. Cells were incubated with the respective cytokine for 40 h at 37 °C, followed by surface and intracellular staining with the Foxp3 Fix/Perm Buffer kit.

**Reporting summary**. Further information on experimental design is available in the Nature Research Reporting Summary linked to this article.

## Data availability
RNA microarray data, RNA sequencing data and ATAC-sequencing data that support the findings of the study have been deposited in the Gene Expression Omnibus (https://www.ncbi.nlm.nih.gov/geo/query/acc.cgi) under the accession number GSE119169. The source data underlying Figs. 1a–d, 2a–c, 3a, b, 3d–f, 4a, c, d, f–j, 5b, d–e, g–j, 6a–h, 7a–d, 8a–d, i, 9b–f, and Supplementary Figs. 1–4 and 8,a, b are provided as a Source Data file. All other data are available from the author upon reasonable request.

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

## Acknowledgements

We thank Alexander Rudensky (Memorial Sloan-Kettering Cancer Center) for providing mice, Frank Lyko for help with amplicon sequencing, Claudia Schmitt, Ulrike Rothermel, Sabine Schmitt, Anna von Landenberg, Kristin Lobbes (all DKFZ), Marina Wuttke, Brigitte Ruhland, Veronika Hofmann, Kathrin Schambeck, Luise Eder, Rudolf Jung (all University Regensburg) and Marie-Luise Brunn (BNITM Hamburg) for excellent technical support. Furthermore, we thank the DKFZ core facilities for preclinical research, microscopy, flow cytometry and genomics & proteomics for outstanding support. This work was supported by grants from the Helmholtz Association of German Research Centers (HZ-NG-505) and the European Research Council (ERC-CoG, #648145 REGiREG) and the Deutsche Forschungsgemeinschaft (DFG, German Research Foundation- Projektnummer 324392634-TRR 221) to M.F., M.D. was supported by the German–Israeli Helmholtz Research School in Cancer Biology.

## Author contributions

M.D., M.B., J.A. and M.F. designed experiments; M.D., Y.H., W.H., F.B., D.K., U.T., A.-C.H., S.B., D.W., A.B. and G.F. performed the experiments; R.M.S. provided material; M.D., C.S., M.B., W.H., C.I., A. H.-W., T.H., H.J.G., M.R., J.A. and M.F. analyzed data; M.D. and M.F. wrote the manuscript.

## Additional information

**Competing interests:** The authors declare no competing interests.

