## [Peer Review File · Nature Communications]

Reviewers' comments:

Reviewer #1 (Remarks to the Author):

The authors identify a new role for RBPj in Treg-mediated T-helper responses. The authors do a thorough and rigorous job of characterizing the phenotype of Treg-specific RBPj deficiency and identify a very intriguing role for RBPj in regulating Treg biology, whereby RBPj specifically restrains a Th2-polarized Treg program. Combined, the data demonstrate a novel role for RBPj and provide important insights into the regulation of Th2 responses by Tregs. The major weakness of the paper is that the authors fail to identify how the altered Treg response in RBPj deficient mice would cause the pathology (splenomegaly, lymphadenopathy, etc.) observed in affected mice. Specific comments are below.

1. In Figure 2 and sup. Fig 3, the authors identify aberrant antibody responses and in Fig. 2f they look at whether or not these antibodies could be binding self-antigens and thus potentially contributing to the pathology in these mice. Considering the overt tissue pathology was restricted to the liver and skin (Figure 1), why did the authors not look at the liver and skin in this panel? And what is the relevance of observing self-antigen responses in organs that did not show any obvious tissue damage?

2. Figure 3e: Bcl6 should be added to the panel to more accurately quantify Tfh frequency/number.

3. Are the data shown in Figure 4J statistically significant?

4. Figure 5a should be changed to have WT and Δ/Δ overlaid on the same graphs (instead of Tconv and Treg) as these need to be directly compared and it is hard to interpret the differences between WT and Δ/Δ in separate graphs. Additionally, the authors suggest that they did not observe any differences between WT and Δ/Δ mice for these markers, however, Gitr does appear to be expressed at higher levels on Δ/Δ Tregs. The MFIs should be quantified and statistical significance determined. If significant, the implications of higher Gitr expression on these cells should be addressed.

-In the results section for Figure 7, the right panel of Figure 7f is referenced when it should be referencing the left panel. Additionally, the right panel is never mentioned in the results section and should be included. In the following paragraph, Figure 7g is discussed but not referenced.

5. Figures 5-8 are all further characterization of the primary phenotype of Treg-specific RBPj-deficiency (expanded "Th2-biased"/"tisTregST2" population) but not an explanation for the cause of pathology in these mice. The paper still leaves the following question unanswered: How does the development of this expanded and altered Th2-biased Treg population fail to suppress Th2 responses and cause disease/pathology in the mice? If anything the KO "Th2-biased" Tregs are either equivalently or even more functional than WT Tregs in many of the assays shown (i.e. functional markers, ex vivo suppressive capacity, frequency/proliferative markers, etc.). The authors suggest the pathology is due to altered Treg function (rather than loss), but how and which of these alterations are causing the pathology are not identified.

Reviewer #2 (Remarks to the Author):

This paper by Delacher and colleagues investigates the role of the transcriptional regulator Rbpj on FOXP3 regulatory T cell function and uncovers a hitherto unknown critical role in regulating their capacity to control Th2 polarization of effector T cells. This function is revealed only in the presence of a suitable environmental cue, probably an infectious agent, explaining the differences

observed in this study compared to those from other researchers. The experimental evidence for this critical role is extremely comprehensive and generally justify the conclusions drawn. However, the large number of results presented do result in a very dense manuscript and at times it is difficult to draw key conclusions.

Responses to Reviewers' Comments (NCOMMS-18-11205-T)

We thank the expert reviewers for their positive opinion, thoughtful comments and scholarly discussion of this manuscript. We have clarified elements of the text and performed extensive additional experimental work. We feel that the manuscript has been substantially strengthened through addressing the comments.

Reviewer #1 (Remarks to the Author):

The authors identify a new role for RBPj in Treg-mediated T-helper responses. The authors do a thorough and rigorous job of characterizing the phenotype of Treg-specific RBPj deficiency and identify a very intriguing role for RBPj in regulating Treg biology, whereby RBPj specifically restrains a Th2-polarized Treg program. Combined, the data demonstrate a novel role for RBPj and provide important insights into the regulation of Th2 responses by Tregs. The major weakness of the paper is that the authors fail to identify how the altered Treg response in RBPj deficient mice would cause the pathology (splenomegaly, lymphadenopathy, etc.) observed in affected mice. Specific comments are below.

We thank the reviewer for these comments and her/his very positive review of this manuscript. We have performed extensive additional experiments to address the altered Treg response in RBPJ deficient mice concerning the inability to control T_H2 responses, which cause the pathology (please see response to point #5).

1. In Figure 2 and sup. Fig 3, the authors identify aberrant antibody responses and in Fig. 2f they look at whether or not these antibodies could be binding self-antigens and thus potentially contributing to the pathology in these mice. Considering the overt tissue pathology was restricted to the liver and skin (Figure 1), why did the authors not look at the liver and skin in this panel? And what is the relevance of observing self-antigen responses in organs that did not show any obvious tissue damage?

We agree with the reviewer that the self-binding properties of the aberrant antibody responses are not entirely clear, especially in regards to the pathology. Therefore, to have a better focus, we moved histology of unaffected organs from Figure 1 to the supplement (Supplementary Figure 2). In addition, we also moved the autoantibody western blots to the Supplement (Supplementary Figure 4). This should avoid confusion, and we thank the reviewer for pointing this out to us.

2. Figure 3e: Bcl6 should be added to the panel to more accurately quantify Tfh frequency/number.

Upon our analysis of Tfh frequency, we did not include the intracellular protein staining for Bcl6. Therefore, we rewrote the figure description (Figure 3e) to clearly indicate that we identify Tfh cells only with the surface markers Cxcr-5 and PD-1. Since the breeding of Rbpj Treg-deficient mice only yields very few homozygous knockout animals, and disease onset cannot be predicted, we would require significant breeding to repeat the Tfh identification with a tripple-staining including Bcl6-, Cxcr-5 and PD-1. Since the presence of Tfh cells is not followed up in the following figures, we voted against breeding many animals to repeat this staining.

3. Are the data shown in Figure 4J statistically significant?

We thank the reviewer for pointing out that we did not include statistics for this plot. We added the the corresponding p-value.

4. Figure 5a should be changed to have WT and Δ/Δ overlaid on the same graphs (instead of Tconv and Treg) as these need to be directly compared and it is hard to interpret the differences between WT and Δ/Δ in separate graphs. Additionally, the authors suggest that they did not observe any differences between WT and Δ/Δ mice for these markers, however, *Gitr* does appear to be expressed at higher levels on Δ/Δ Tregs. The MFIs should be quantified and statistical significance determined. If significant, the implications of higher *Gitr* expression on these cells should be addressed.

We agree with the reviewer and changed the histograms in Figure 5a accordingly. Direct comparison is now much easier. We show the histograms for Foxp3, CTLA-4 and HELIOS expression in Figure 5a. The global comparison between WT and Δ/Δ Treg cells is shown in Figure 6. We thank the reviewer for pointing this out.

-In the results section for Figure 7, the right panel of Figure 7f is referenced when it should be referencing the left panel. Additionally, the right panel is never mentioned in the results section and should be included. In the following paragraph, Figure 7g is discussed but not referenced.

We simplified Figure 7 and corrected all references in the main text. We thank the reviewer for pointing out this labelling error and apologize for the inconvenience.

5. Figures 5-8 are all further characterization of the primary phenotype of Treg-specific RBPj-deficiency (expanded “Th2-biased”/“*tis*TregST2” population) but not an explanation for the cause of pathology in these mice. The paper still leaves the following question unanswered: How does the development of this expanded and altered Th2-biased Treg population fail to suppress Th2 responses and cause disease/pathology in the mice? If anything the KO “Th2-biased” Tregs are either equivalently or even more functional than WT Tregs in many of the assays shown (i.e. functional markers, ex vivo suppressive capacity, frequency/proliferative markers, etc.). The authors suggest the pathology is due to altered Treg function (rather than loss), but how and which of these alterations are causing the pathology are not identified.

We agree with the reviewer and performed multiple experiments to clarify this issue (**new Figure 9**).

Recently, it was shown that *Lilrb4a* (encoding the protein ILT3) expressing Treg cells were unable to regulate TH2-responses due to their inability to control the maturation of a specific TH2-promoting DC subset. This subset of DCs is characterized by the expression of PD-L2 and IRF-4. Our ATAC-seq data identified a highly-accessible region at the *Lilrb4a* promoter in *Klrg1*⁺*ST2*⁺ Treg cells isolated from affected *Rbpj*-

deficient animals. Indeed, enhanced activity at the *Lilrb4a* promoter resulted in increased *Lilrb4a* expression in *Klrg1*⁺*ST2*⁺ Treg cells from affected Δ/Δ animals (**new Figure 9a,b**). To study the link between Gata-3 overexpression, ILT3-expression and control of TH2 responses, we performed in-vitro polarization studies with Treg cells. IL-4 is the prototype cytokine to induce Gata-3 expression and TH2 differentiation, and IL-33 is linked to the generation of *tisTregST2* cells. We were able to co-induce ILT3 and Gata-3 over-expression specifically in the TH2 polarized IL-4 and IL-33 treated Treg cells. Using this model, we studied the ability of ILT3-expressing TH2-polarized Treg cells to influence DC maturation and DC-mediated TH2 polarization of naive T cells in-vitro. Our data revealed that ILT3-expressing TH2-polarized Treg cells profoundly promoted the differentiation of PD-L2⁺IRF4⁺ DCs, a subset described to support TH2 polarization in vivo (**new Figure 9d**). In addition, ILT3-expressing Treg cells were unable to suppress the TH2 differentiation of IL-4 and CD3-stimulated naive T cells, in the presence of DCs, into Gata-3-polarized effector T cells (**new Figure 9e**). These new data strongly indicate that overexpression of Gata-3 and ILT3 renders Treg cells unable to properly suppress TH2 responses.

Therefore, our findings challenge the current dogma that TH2-polarized Gata-3^{high} Treg cells are better suppressors of the corresponding TH2-polarized effector T cells, a model proposed based on the complete deficiency of IRF4.

The elevated sensitivity of *Rbpj*-deficient Treg cells towards Gata-3 induction by IL-4 can explain the profound expansion of Gata-3^{high}*Klrg1*⁺*ST2*⁺ TH2-polarized Treg cells in affected Δ/Δ animals, with ameliorated TH2-suppressive potential, which results in the observed TH2 pathology. The failure of ILT3-expressing Treg cells to suppress TH2 responses has been published recently (Ulges, A. *et al.* Protein kinase CK2 enables regulatory T cells to suppress excessive TH2 responses in vivo. *Nat Immunol* **16**, 267-275 (2015)) and supports our findings.

Corresponding changes in the manuscript text can be found in the results and discussion section:

p. 16-18 and p. 20-21

Reviewer #2 (Remarks to the Author):

This paper by Delacher and colleagues investigates the role of the transcriptional regulator Rbpj on FOXP3 regulatory T cell function and uncovers a hitherto unknown critical role in regulating their capacity to control Th2 polarization of effector T cells. This function is revealed only in the presence of a suitable environmental cue, probably an infectious agent, explaining the differences observed in this study compared to those from other researchers. The experimental evidence for this critical role is extremely comprehensive and generally justify the conclusions drawn. However, the large number of results presented do result in a very dense manuscript and at times it is difficult to draw key conclusions.

We thank the reviewer for these very positive remarks. To clarify the manuscript and reduce the number of results in the main figures, we changed the layout as follows: we moved histology of unaffected organs from Figure 1 to now Supplementary Fig 2; we also removed organ weights and liver enzymes. We moved autoantibody western blots from Figure 2 to now Supplementary Figure 4. We removed IL-4 expression in

Tfh cells in Figure 3. We removed some histograms in Figure 5. We focused the data in Figure 7. We moved parts of Figure 8 to new Figure 9 and merged it with experiments requested by reviewer 1. In the text, we shortened paragraphs and added more summary sentences. We tried to clarify the message in the discussion. We thank the reviewer for pointing this out and we hope that she/he finds the manuscript now improved in clarity and less dense.

Reviewers' comments:

Reviewer #1 (Remarks to the Author):

The authors identify a new role for RBPj in Treg-mediated T-helper responses. The authors do a thorough and rigorous job of characterizing the phenotype of Treg-specific RBPj deficiency and identify a very intriguing role for RBPj in regulating Treg biology, whereby RBPj specifically restrains a Th2-polarized Treg program. Combined, the data demonstrate a novel role for RBPj and provide important insight into the regulation of Th2 responses by Tregs. The one weakness from the original submission was that the authors failed to identify how the altered Treg response in RBPj deficient mice would cause the pathology (splenomegaly, lymphadenopathy, etc.) observed in affected mice. In the revised manuscript, the authors provide data suggesting that the pathology may be due to a defect in the ability of the transgenic Tregs to suppress Th2-promoting DCs via expression of the protein ILT3. While the new data are intriguing there are a few remaining concerns outlined below:

1. If the lack of Treg-suppression on Th2 responses leads to sufficiently elevated Th2 responses to result in the spontaneous pathology, it is a bit concerning that the parasitic burden in the Th2/parasite model were not significantly different between control and transgenic mice (Figure 4J).
2. At multiple points the authors claim that the Th2-polarized Tregs are less suppressive. However, in the previous submission, Figure 5C contained data showing that the in vitro suppressive capacity of the Rbpj-deficient Tregs was not impaired. This panel has been removed from the paper in the current submission. While this is not a direct contradiction, as I believe the authors are intending to suggest that these cells are defective specifically in their capacity to suppress Th2 differentiation, the authors need to justify why these data were removed as they are still relevant to understanding the functional capacity of these cells.
3. The authors make the claim that the new data "change the dogma" of how Th2-polarized Treg cells function. While possible, this is an overinterpretation of the data which are still largely correlative; the authors never definitely prove that ILT3 expression is the cause of the functional defects and pathology in the RBPj-deficient mice. This may be outside the scope of the current story, but needs to be more thoughtfully discussed given points 1 and 2 above, and due to the lack of definitive causation.

Reviewer #3 (Remarks to the Author):

I find this a very interesting paper and the amount of data provided is staggering.

With regard to the response to reviewer 1:

Most of the comments were addressed adequately or well enough, with the exception of comment 5, in my view.

Comment 2. The reviewer asked for Bcl6 staining to properly analyze Tfh cells. The authors argue that it would take a very large effort to provide this staining and I agree that the expected gain from doing this is not sufficient to justify this big effort. That said, it is a pity that the Tfh cells were not followed up on. Clearly, there is something major happening there, as indicated by the prominent effects on specific Ig levels. On the other hand, the authors already do a lot, so I think this is forgivable.

Comment 4. The authors did ignore a request to show MFI statistics. Again, I think this is forgivable, as the point they are making is that expression of these markers is unaltered. They

apparently took out the GITR staining that must have been here in an earlier version of the manuscript without justification. Again, this seems a minor point, however.

Comment 5. The reviewer demanded a mechanistic explanation for the pathology in mice with Treg specific deficiency for RBPJ. I am torn about the response by the authors. They clearly made a big effort, and report some interesting new findings, but the mechanism they now propose is not well worked out. They argue that RBPJ normally inhibits the development of ILT3+ tissue type Tregs and that these Tregs promote the differentiation of PD-L2+IRF4+ DC, which in turn induce Th2 responses. The authors' interpretation is that by having more of these ILT3+ Tregs in the knock out mice, the downstream cascade to Th2 responses would logically be enhanced. The experiments shown do not however directly address this possibility. No evidence is provided that there are indeed more ILT3+ Tregs in the knock out mice. In fact, the link to ILT3 is made on the basis of ATACseq and RNAseq data, in which KLRG1- (!) wild type Tregs are compared to KLRG1+ (!) knock out Tregs, a comparison with two variables. Is upregulation of ILT3 due to RBPJ deficiency or merely associated with development into KLRG1+ cells? Although either answer would be fine, providing the proper samples for the comparison is necessary. It does seem important, moreover, to show that there actually are more ILT3+ Tregs in these knock out mice. Furthermore, the propensity of ILT3+ Tregs to promote the development (in vitro) of PDL2+IRF4+ DC is shown for wild type ILT3+ Tregs, made in vitro. No evidence is shown however that this is what RBPJ deficient Tregs do using the same type of in vitro setting (but without pretreating the knock out Tregs with IL4 and IL33). Also, are there more PD-L2+IRF4+ DC in affected ko mice? Likewise, do the knock out Tregs not suppress Th2 responses in vitro, as shown for ILT3+ wild type Tregs in Figure 9E? Are the DC actually required in this setting? Do the Tregs do anything here at all (suppress proliferation)? In short, the argument made is too dependent on extrapolation indirect evidence and requires experiments that more directly address the role of RBPJ in all this.

Minor point: I cannot see the dots in the dot plot in Figure 9d.

Specific responses to the reviews comments:

We thank the expert reviewers for their positive opinion, thoughtful comments and scholarly discussion of this manuscript.

Reviewer 1:

Comment 1:

If the lack of Treg-suppression on Th2 responses leads to sufficiently elevated Th2 responses to result in the spontaneous pathology, it is a bit concerning that the parasitic burden in the Th2/parasite model were not significantly different between control and transgenic mice (Figure 4J).

Our response: We do find a trend towards reduced numbers of parasites in the intestine ($p=0.079$) in the Rbpj-deficient animals at day 6 (Figure 4 j). And we also find significantly reduced larval output of *S. ratti* DNA in Rbpj-deficient animals at this time point (Figure 4 i). In general, the experiment was performed with five animals per group and our time points were chosen according to our primary focus, the induction of Gata-3 positive Treg cells and Th2 effector cells by the parasites.

Comment 2: At multiple points the authors claim that the Th2-polarized Tregs are less suppressive. However, in the previous submission, Figure 5C contained data showing that the *in vitro* suppressive capacity of the Rbpj-deficient Tregs was not impaired. This panel has been removed from the paper in the current submission. While this is not a direct contradiction, as I believe the authors are intending to suggest that these cells are defective specifically in their capacity to suppress Th2 differentiation, the authors need to justify why these data were removed as they are still relevant to understanding the functional capacity of these cells.

Our response: In the revision we removed the T-helper subtype-unspecific *in-vitro* suppression data from Figure 5C and added a much more detailed TH2-specific suppression assay in Figure 9E. Now, we re-introduced the *in-vitro* suppression data from the standard Treg suppression assay in Supplementary Figure 3 and mentioned it in the text (line 111 – 114; “.....indicating that Rbpj deficiency in Treg cells did not lead to a global loss of Treg-mediated immune control. This was supported by data from a standard *in vitro* suppression assay with TCR-stimulated T responder cells, where we did not detect significant changes in the *in-vitro* suppressive potential (Supplementary Figure 3). In summary, these data indicate that RBPJ deficiency affected a more specific segment of Treg function.”)

Comment 3: The authors make the claim that the new data “change the dogma” of how Th2-polarized Treg cells function. While possible, this is an overinterpretation of the data which are still largely correlative; the authors never definitely prove that ILT3 expression is the cause of the functional defects and pathology in the RBPj-deficient mice. This may be outside the scope of the current story, but needs to be more thoughtfully discussed given points 1 and 2 above, and due to the lack of definitive causation.

Our response: We agree with the reviewer that the identification of the direct link between ILT3 and the functional defects and pathology in Rbpj-deficient animals would go well beyond the scope of this manuscript. The causative link between ILT3 and TH2-suppression have already been shown and discussed elsewhere (Ulges A. *et al.*, “Protein kinase CK2 enables regulatory T cells to suppress excessive TH2 responses *in vivo*”, *Nat Immunol* 2015; Bird L.,

“Regulatory T cells: CK2: keeping TH2 cells in check”, Nat Rev Immunol 2015). Still, we agree that the causative link between ILT3 and the pathology in Rbpj-deficient mice is not definitive. Therefore, we carefully toned down the interpretation in the abstract, result and discussion part.

Reviewer 3:

Comment 2. The reviewer asked for Bcl6 staining to properly analyze Tfh cells. The authors argue that it would take a very large effort to provide this staining and I agree that the expected gain from doing this is not sufficient to justify this big effort. That said, it is a pity that the Tfh cells were not followed up on. Clearly, there is something major happening there, as indicated by the prominent effects on specific Ig levels. On the other hand, the authors already do a lot, so I think this is forgivable.

Our response: We thank the reviewer that she/he agrees with our strategy to avoid disproportionate animal experimentation to gain information that is not followed up in the course of the manuscript.

Comment 4. The authors did ignore a request to show MFI statistics. Again, I think this is forgivable, as the point they are making is that expression of these markers is unaltered. They apparently took out the GITR staining that must have been here in an earlier version of the manuscript without justification. Again, this seems a minor point, however.

Our response: Reviewer 1 asked us to modify Figure 5 during revision 1 (*“Figure 5a should be changed to have WT and Δ/Δ overlaid on the same graphs (instead of Tconv and Treg) as these need to be directly compared and it is hard to interpret the differences between WT and Δ/Δ in separate graphs...”*). We changed the graph according to the reviewers wishes. In addition, reviewer 2 claimed that the manuscript be too dense (*“However, the large number of results presented do result in a very dense manuscript and at times it is difficult to draw key conclusions.”*). In accordance with both reviewers wishes, we simplified the figure and removed parts which did not show significant alterations or were redundant since we show a complete gene expression profile of Rbpj-KO vs WT Treg cells in Figure 6 – this figure recapitulates the flow cytometry data from Figure 5A.

Comment 5. The reviewer demanded a mechanistic explanation for the pathology in mice with Treg specific deficiency for RBPJ. I am torn about the response by the authors. They clearly made a big effort, and report some interesting new findings, but the mechanism they now propose is not well worked out. They argue that RBPJ normally inhibits the development of ILT3+ tissue type Tregs and that these Tregs promote the differentiation of PD-L2+IRF4+ DC, which in turn induce Th2 responses. The authors' interpretation is that by having more of these ILT3+ Tregs in the knock out mice, the downstream cascade to Th2 responses would logically be enhanced. The experiments shown do not however directly address this possibility. No evidence is provided that there are indeed more ILT3+ Tregs in the knock out mice. In fact, the link to ILT3 is made on the basis of ATACseq and RNAseq data, in which KLRG1- (!) wild type Tregs are compared to KLRG1+ (!) knock out Tregs, a comparison with two variables. Is upregulation of ILT3 due to RBPJ deficiency or merely associated with development into KLRG1+ cells? Although either answer would be fine, providing the proper samples for the comparison is necessary. It does seem important, moreover, to show that there actually are more ILT3+ Tregs in these knock out mice. Furthermore, the propensity of ILT3+ Tregs to promote the development (in vitro) of PDL2+IRF4+ DC is shown for wild type ILT3+ Tregs, made in vitro. No evidence is shown however that this is what RBPJ deficient Tregs do using the same type of in vitro setting (but without pretreating the knock out Tregs with IL4 and IL33). Also,

are there more PD-L2+IRF4+ DC in affected ko mice? Likewise, do the knock out Tregs not suppress Th2 responses in vitro, as shown for ILT3+ wild type Tregs in Figure 9E? Are the DC actually required in this setting? Do the Tregs do anything here at all (suppress proliferation)? In short, the argument made is too dependent on extrapolation indirect evidence and requires experiments that more directly address the role of RBPJ in all this.

Our response: We thank reviewer 3 for pointing out that we made a big effort and report some interesting new findings. In our efforts to reduce disproportionate animal experimentation, especially in the case of $\text{Foxp3}^{\text{Cre}}\text{Rbpj}^{\text{fl/fl}}$ animals, where breeding is only possible in het/het mode, we decided to answer the initial reviewer's questions with extensive in-vitro experimentation as well as further exploration of our already established datasets. We argued that $\text{Klrg1}^+\text{ST2}^+$ Treg cells from KO animals expressed more ILT3 than $\text{Klrg1}^-\text{ST2}^-$ Treg cells from WT animals (old Figure 9B). We agree with reviewer 3 and added two more groups to our new Figure 9B, to now include WT $\text{Klrg1}^+\text{ST2}^+$ Treg cells as well. This dataset indicates that, also in WT animals, $\text{Klrg1}^+\text{ST2}^+$ Treg cells expressed enhanced ILT3. These data suggest that upregulation of ILT3 is not due to RBPJ deficiency and is more likely associated with development into $\text{Klrg1}^+\text{ST2}^+$ Treg cells in general. We discussed these additional data in the text (line 369-372) and thank the reviewer for this suggestion. In addition, we carefully revised the text to avoid overinterpretation of our *in-vitro* generated datasets. For more details, please consider our direct response to reviewer 1. We thank reviewer 3 for her/his fast review of our responses to reviewer 1.

Minor point: I cannot see the dots in the dot plot in Figure 9d.

Our response: We thank the reviewer for this important comment. Even though the dots were visible on our PDF document, we changed the graph type to a contour plot with outliers (a style accepted by the guidelines of Nature journals) to enhance the visibility of our populations.

REVIEWERS' COMMENTS:

Reviewer #3 (Remarks to the Author):

I have no further comments
Derk Amsen